# The three stages of religious decline around the world

Jörg Stolz [1,2] ✉, Nan Dirk de Graaf [3,4], Conrad Hackett [5,6] &
Jean-Philippe Antonietti [7]

Religiosity tends to decline across generations. However, religious decline is more pronounced in some countries and the diminishing aspects of religion vary by context. To explain such variation, we extend the general secular transition model, which proposes that countries undergo a similar process of secularization beginning at different points in time. We explain that secular transition happens in three steps: first, public ritual participation declines; second, the importance of religion to individuals declines; and third, people shed religious affiliation. We test this model using datasets from surveys in 111 countries (Pew Research Center), 58 countries (World Values Survey and European Values Study (WVS/EVS)), and a subset of 17 countries measured in at least five WVS/EVS waves. We show the model fits countries with Christian, Muslim, Hindu, and Buddhist pluralities. While Eastern post-Soviet countries deviate from this pattern, traditionally Muslim countries appear to follow its early stages. However, we recommend caution in interpreting longitudinal claims, due to limited data.

It is well established that religion declines in Western countries. The secular transition model provides a prominent theoretical explanation for this phenomenon[1,2]. It suggests that religiosity declines over roughly 200 years, mirroring the demographic transition. The model links modernization, particularly institutional and technological innovation, to a decline in the symbolic and social functions of religion[3,4]. As secular institutions offer more efficient solutions to life's problems —e.g., biomedicine replacing religious healing, welfare states replacing spiritual reassurance—religion loses functional relevance. This transformation is assumed to occur not primarily by individuals abandoning religion during their adult life, but by an intergenerational breakdown of religious socialization.

The model of the secular transition is compatible with ideas from evolutionary anthropology, specifically mechanisms involving credibility enhancing displays (CRED's)[5]. The central idea of CRED's is that individuals will believe and imitate others with a higher probability if these others back up their words with hard-to-fake deeds[6]. Especially in religion, where supernatural entities are notoriously invisible, CRED's are thought to be of primary importance[7,8]. The link of this insight to our model is as follows: Because of the increasing efficiency of secular goods, parents may still teach their children religiosity—but stop backing it up with religious behavior. This will lead children to drop their religiosity levels.

Previous studies testing this model have focused on western countries that happen to have Christian majorities. These studies have confirmed that different western countries enter the secular transition at different points in time and that aggregate religiosity declines via cohort replacement[9–14]

It has been unclear whether the model of the secular transition can be extended to the non-western world and to countries with non-Christian majorities[15–17]. While it has been shown that younger generations in most countries around the world tend to be less religious

[1]Institute for the Social Sciences of Religions, Faculty of Theology and Religious Studies, University of Lausanne, Lausanne, Switzerland. [2]LIVES. Swiss Centre of Expertise on Life-Course Research, University of Lausanne, Lausanne, Switzerland. [3]Nuffield College, University of Oxford, Oxford, UK. [4]Department of Sociology, University of Oxford, Oxford, UK. [5]Pew Research Center, Washington, DC, USA. [6]Maryland Population Research Center, University of Maryland, College Park, MD, USA. [7]Institute of Psychology, Faculty of Social and Political Sciences, University of Lausanne, Lausanne, Switzerland. ✉e-mail: joerg.stolz@unil.ch

than older generations, these differences do not show up reliably with all indicators of religiosity. For instance, Senegal shows age gaps in participation in worship services but no similar age gaps in the importance of religion or belonging (identifying with a religion). The Netherlands has age gaps in belonging only, Portugal across all three, and countries like Tanzania or Kazakhstan show no marked cohort differences in any of these indicators[18]. It has remained a puzzle, why this should be the case.

Here, we show that the puzzle of why cohort differences do not show up reliably in the same indicators across countries can be explained by extending the secular transition model to a global context and specifying the process through which religiosity declines across its different dimensions. We propose that secularization follows a consistent sequence: first, participation in public rituals declines; second, importance of religion drops, and third, people shed their formal belonging. We refer to this as the Participation–Importance–Belonging (P-I-B) sequence. This ordering reflects the relative costliness of each behavior: public ritual attendance demands the most time and energy, while mere identity affiliation is least burdensome and thus most persistent. The P-I-B sequence normally does not occur within the lifespan of an individual but rather transpires across extended periods among cohort groups. Using Pew surveys (2008–2023) and seven waves of the World/European Values Survey (WVS/EVS) (1981–2020), we analyze cohort differences in three dimensions of religiosity–public ritual participation, private importance, and religious belonging–in more than 100 countries across major religious traditions. In line with our expectations, we find that countries with high religiosity show cohort differences mainly in participation, countries with intermediate religiosity have differences across all three dimensions, and highly secular countries vary primarily in belonging. We replicate the finding with wave 7 of the WVS/EVS dataset, which includes 58 countries. We also test a longitudinal version of this hypothesis using change over time on 17 countries tracked over at least five waves. In all three tests, the PIB sequence can be observed, although the longitudinal findings remain limited by the time span of available data and because the secular transition is at an early stage in some countries.

## Results

### Three stages of religious decline

Our initial examination focuses on the countries in the Pew dataset. We investigate whether the degree of secularity of the country covaries with age-gaps in participation, importance, and belonging, as hypothesized above.

The central independent variable country secularity is the first principal component of a principal component analysis of the three standardized variables covering percentages of individuals with weekly attendance, finding religion very important in their lives, and being affiliated with a religion. This component represents 88% of the variability. With such a high value for the first component, it is probable that the remaining components carry little information. We can therefore restrict our analysis to this first dimension.

In a next step, we rank the countries according to country secularity (top: most religious; bottom: most secular) and plot the age gap differences between individuals 40+ and under 40 for participation, importance, and belonging (Fig. 1). We can interpret this ranking as the ordinal position of the countries on the secular transition. Positive differences show younger individuals being less religious than older individuals. We observe that most differences are positive; this means that, as expected, in most countries younger people are less religious than older people.

We see that participation differences are most important at the beginning of the secular transition (blue, at the top of the graph), importance differences take an intermediate position (green, in the middle of the graph), and belonging differences appear mainly at the end of the transition (red, at the bottom of the graph). For example,

cohort-religiosity differences in Senegal (a country at the beginning of the secular transition, 13th from top) all concern participation, whereas cohort-religiosity differences in Denmark (a country close to the end of the secular transition, 5th from bottom) concern overwhelmingly belonging.

The most religious countries, such as Ethiopia, Nigeria, Niger, and Mali show the lowest or even reversed cohort-religiosity differences. This fits the expectation of the secular transition model which assumes that these countries have not yet entered the secular transition.

For different reasons, Israel also clearly diverges from the predictions of the model. Israel is exceptional in many analyses of worldwide religion. It is the only nation where Jews are a majority of the population. The modern state of Israel is a relatively young political entity, initially shaped by many secular migrants. Younger Israelis are increasingly the offspring of very religiously conservative Jews, who have more children than more secular Jews. Israel also has strong external and internal tensions with Muslim populations. Because of such factors, religion has a high degree of salience in the lives of many Israelis[19].

In the next step, we seek to model the relationships we have just observed. To do so, we cast the indicators participation, importance, and belonging as a sigmoid (inverse logit) function of country secularity both for older (40 + ) and younger individuals (18–39) with

$$RI = \left(1 + e^{(1-old)*a_1(CS-b_1-BX) + old*a_2(CS-b_2-BX)}\right)^{-1} \quad (1)$$

where RI = religiosity indicator; CS = country secularity; old = dummy variable for being older (40 + ); $a_1$ = slope for younger individuals; $b_1$ = mid-point for younger individuals; $a_2$ = slope for older individuals; $b_2$ = mid-point for older individuals; $B$ = vector for estimates of control variables; $X$ = vector of control variables.

In Fig. 2a, we plot the predicted values of the estimated functions for participation, importance, and belonging as a function of country secularity for older (red) and younger (green) individuals. We also plot the difference between the predicted values for older and younger, giving the blue line. Vertical dashed lines show the midpoints of the functions for older and younger individuals combined. We see that younger individuals are on all measures less religious than older individuals, but that the location of the indicators varies with respect to country secularity. Going from low to high country-secularity, participation declines earlier than importance, belonging declines last.

In Fig. 2b, we only extract the differences between older and younger (the blue lines in Fig. 2a) and separately graph them. Looking only at these differences between older and younger individuals, we see a rise and fall of participation-differences happening earlier than a rise and fall of importance-differences, and lastly a rise and fall of belonging-differences. The assumption we are making here is that country secularity may be interpreted in a temporal way, that is, that countries with a higher religiosity (lower country secularity) over time move towards more secularity and show similar declines first in participation, then in importance, and lastly in belonging.

We calculate the difference between the midpoints of older (red in Fig. 2a) and younger (green in Fig. 2a) cohorts for participation, importance, and belonging. Using 95% highest posterior density intervals (HPDIs), we find that all differences are significantly below zero: participation: −0.162 [−0.252, −0.072]; importance: −0.164 [−0.203, −0.126]; belonging: −0.196 [−0.325, −0.080]. These results indicate robust generational declines across all three dimensions.

We also examine within-cohort contrasts between indicators by calculating the difference between midpoints of the difference functions (e.g., participation minus importance) for younger and older cohorts separately. All contrasts are significantly different from zero, suggesting distinct relational structures among indicators within each cohort. For younger cohorts, the differences are: participation –

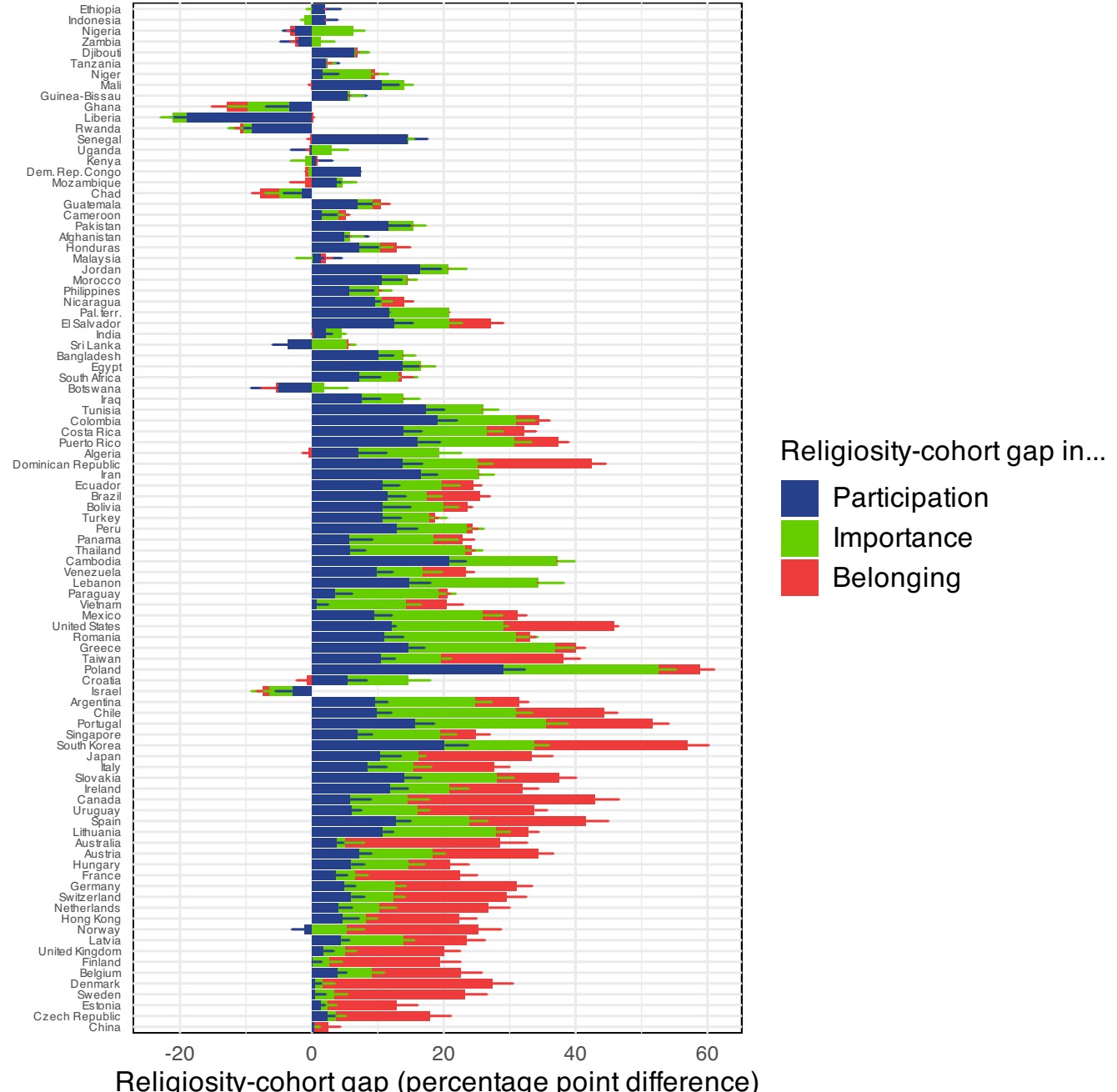

**Fig. 1 | Religiosity-cohort gap in participation, importance, and belonging in different countries (country secularity ranked).** Pew data. All countries shown, except eastern post-communist countries. Differences in percentage of participation, importance, and belonging between younger (< 40) and older (+ 40) individuals in different countries. Positive differences show younger individuals being less religious than older individuals. The countries on the y axis are ranked from most religious (top) to least religious (bottom). We plot the 95% confidence intervals. For better visibility, only half of the confidence interval is plotted - for positive percentage values at the right of the respective percentage bar, for negative percentage values at the left of the respective percentage bar.

importance: −0.383 [−0.453, −0.315]; participation − belonging: −1.346 [−1.455, −1.244]; importance − belonging: −0.962 [−1.050, −0.885]. For older cohorts, we find: participation − importance: −0.386 [−0.460, −0.312]; participation − belonging: −1.380 [−1.520, −1.258]; importance − belonging: −0.994 [−1.117, −0.893].

### Religious decline on different continents
The secular transition and the P-I-B sequence appear to be underway across continents.

However, the continents are in different temporal stages of the overall P-I-B process. African countries are in the beginning, the

Americas and Asia & Oceania in the middle, while Europe is at a later stage (Fig. 3). Africa is in the first stage of the secular transition and only shows the first two steps of the P-I-B sequence (Supplementary Methods 1). These two steps, however, are clearly identifiable and the differences between indicators are statistically significant. For younger cohorts in Africa, we observe: participation − importance: −0.355 [−0.533; −0.212]; for older cohorts, we find: participation − importance: −0.522 [−0.982; −0.229]. The generational differences between younger and older respondents in each of the two indicators participation: −0.136 [−0.298; −0.002], and importance: −0.303 [−0.768; −0.001], however, do not turn out to be significant.

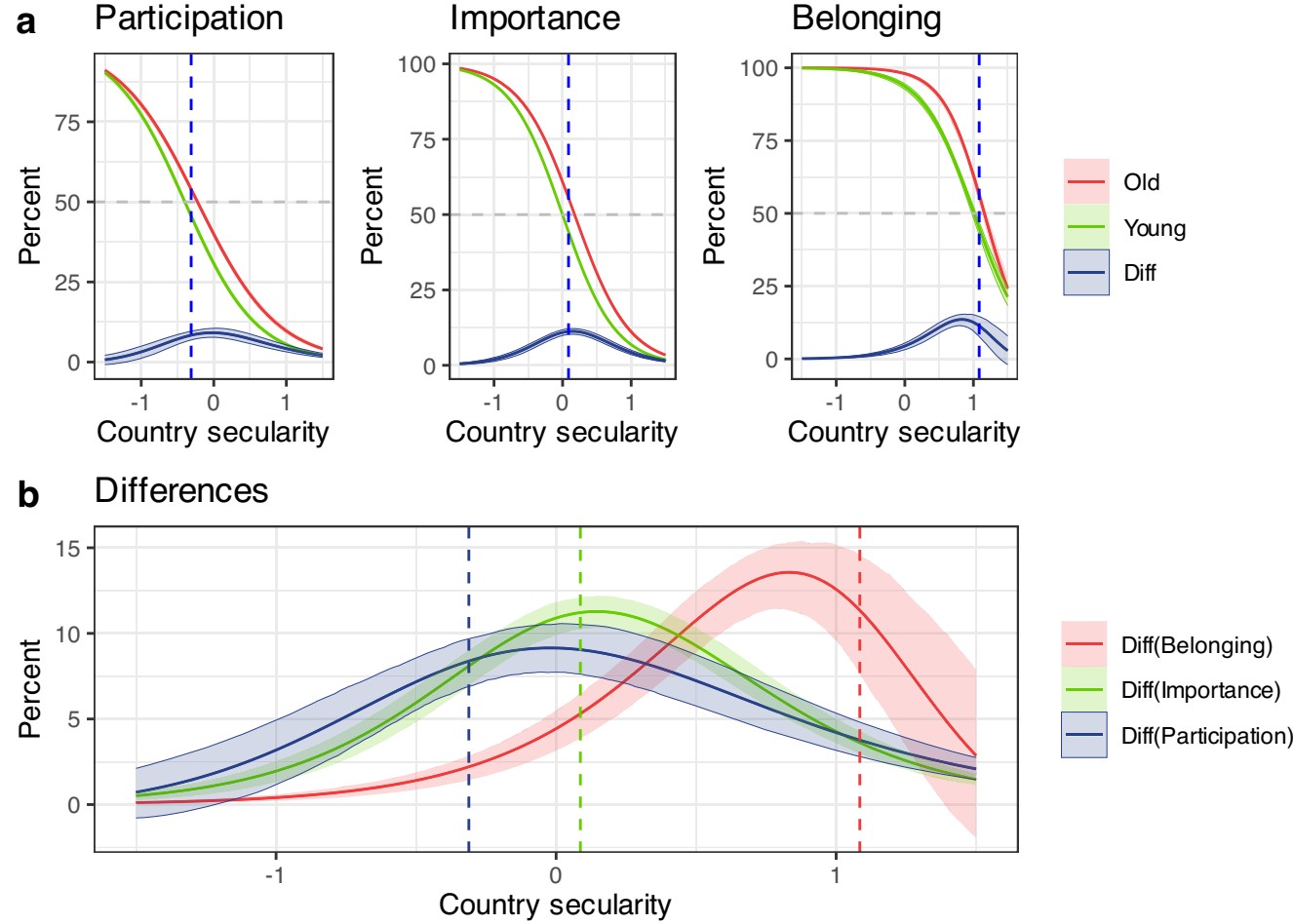

**Fig. 2 | Country secularity and religiosity-cohort gap in participation, importance, and belonging.** Pew data. Based on all countries. Diff = difference between older and younger cohorts. **a** shows predicted values of the estimated functions for participation, importance, and belonging as a sigmoid (inverse logit) function of country secularity for older (40 + , red) and younger (< 40, green) individuals. The blue line is the difference between predicted values for older and younger individuals. Dotted vertical lines in blue show the midpoints of the estimated function for participation, importance, and belonging for the overall population. We show 50% error bands (credible intervals) around the lines. **b** shows the rise and fall of participation-, importance-, and belonging differences between older and younger individuals in one graph. These are the same curves as depicted in blue in **a**, but here we present them concisely in one Figure. We show 50% error bands (credible intervals) around the lines. The dotted vertical lines show the midpoints of the estimated functions for participation, importance, and belonging (corresponding to the vertical dotted blue lines in **a**.

That having been said, African countries are still very religious and there is no African country approaching the middle or the end of the secular transition. Thus, only the future can tell whether they will take the third step, as predicted by our model. All European countries are in medium or later stages of the secular transition; for these countries, both our statistical and historical evidence do indeed suggest a P-I-B trajectory[20]. In these countries, participation started to drop in the 19th century, the drop in finding religion important followed subsequently, and the drop in religious belonging started in the 1960s.

**Religious decline in countries with different historical religions**
The secular transition and the P-I-B sequence manifest in countries with traditionally Christian, Buddhist/Hindu, and Muslim backgrounds, although traditionally Muslim countries only exhibit the first two stages (P-I) (Fig. 4 and Supplementary Methods 2). In 69% of the world's countries and territories, Christians are the largest religious group[21]. Accordingly, most countries in our datasets are traditionally Christian, and these countries span the range of the secular transition. Here, the P-I-B sequence is most clearly visible. While our dataset includes fewer traditionally Buddhist and Hindu countries, the P-I-B

sequence remains discernible. We acknowledge that participation in acts of worship at temples within Asian religions differs from participation in congregational religions. For this reason, we employed a specialized measure tailored to East Asian societies in order to accurately assess participation in that region (see the methods section under dependent variables below). Notably, when using data specifically designed to measure participation at Buddhist sites, we observe the same age-related patterns in attendance that are evident in religions emphasizing congregational worship. This is particularly remarkable given that some scholars have argued that concepts such as participation or belonging are not or only with difficulty applicable to non-congregational Asian religions such as Buddhism or Hinduism[22–24].

In the case of traditionally Muslim countries, we find that they are all situated in the first stages of the secular transition and only show the first two steps of the P-I-B sequence. These two steps, however, are clearly identifiable and their difference is statistically significant. For younger cohorts, we observe: participation – importance: −0.536 [−0.612; −0.46]; For older cohorts, we find: participation – importance: −0.506 [−0.586; −0.424]. The generational differences between younger and older respondents in each of the two indicators

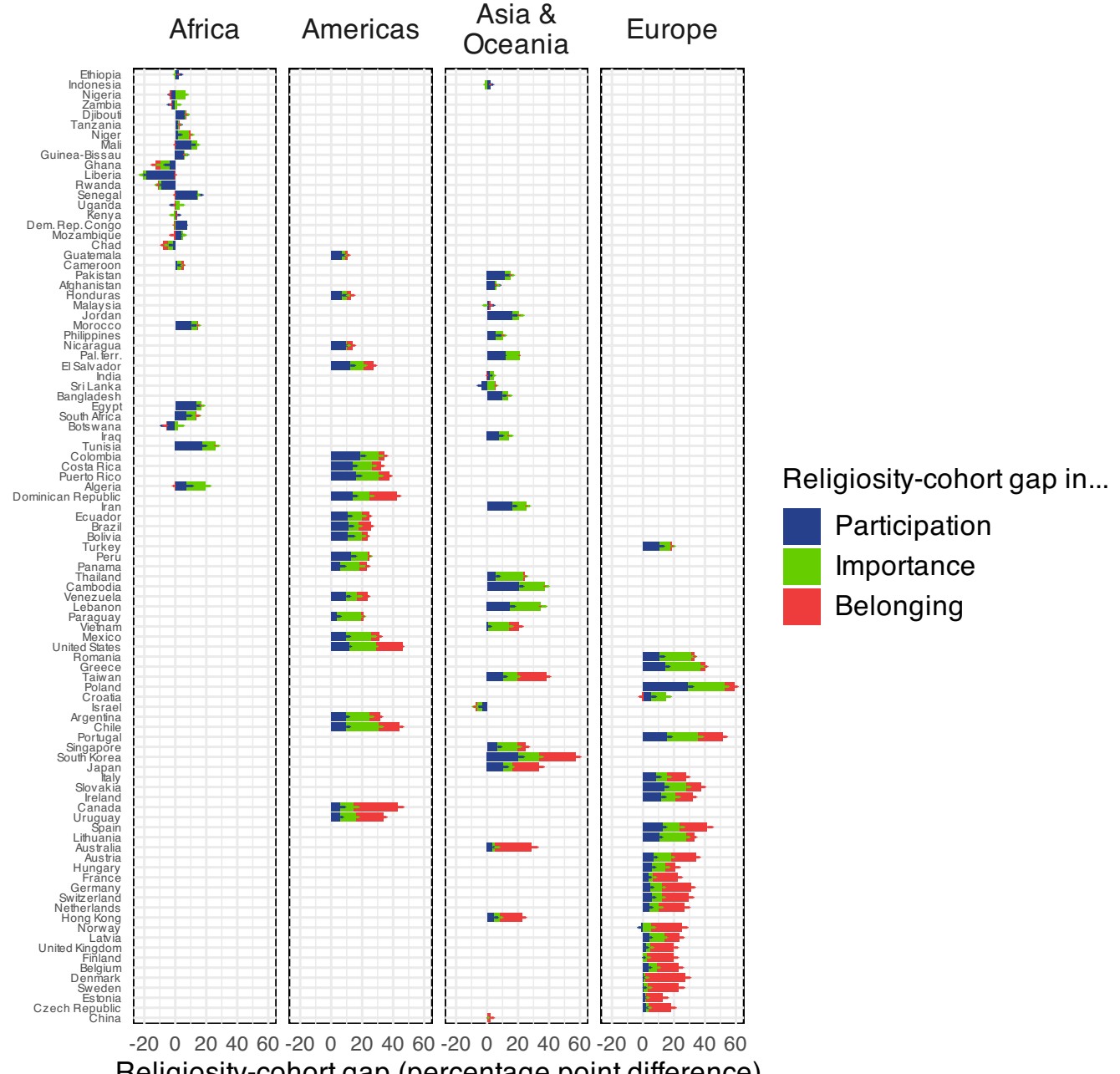

**Fig. 3 | Religiosity-cohort gap in participation, importance, and belonging in different countries and continents (country secularity ranked).** Pew data. All countries shown, except eastern post-communist countries. Differences in percentage of weekly attendance, high importance of religion and religious belonging between younger (<40) and older (+40) individuals in different countries. Positive differences show younger individuals being less religious than older individuals. The countries on the y axis are ranked from most religious (top) to least religious (bottom). We plot the 95% confidence intervals. For better visibility, only half of the confidence interval is plotted - for positive percentage values at the right of the respective percentage bar, for negative percentage values at the left of the respective percentage bar.

participation: −0.171 [−0.254; −0.09], and importance: 0.141 [−0.218; −0.067] are equally significant.

Due to limits of data and the current status of the secular transition in countries with non-Christian religious pluralities, we cannot be certain that these countries will follow the stages predicted by our model. The number of traditionally Buddhist and Hindu countries in our dataset is small. Consider, for example, that the only countries in the world where Hindus are the largest group are Nepal, India, and Mauritius. The only country with a Jewish majority is Israel; with respect to the secular transition, this country seems to be, as already noted, an outlier.

## The P-I-B sequence in post-communist countries

In our in-depth analysis, we find that the P-I-B sequence does not seem to apply to eastern post-communist countries (Fig. 5). These countries have a communist past and have traditionally Orthodox or Muslim majorities. Examples are Russia, Serbia, Georgia, or Tajikistan. Cohort differences are small for all three types of measures and are sometimes negative (with younger individuals being more religious than older individuals). By contrast, western post-communist countries are traditionally Protestant or Catholic. Examples are Poland, Lithuania, Slovakia, and the Czech Republic. These countries conform relatively well to the predictions of the model.

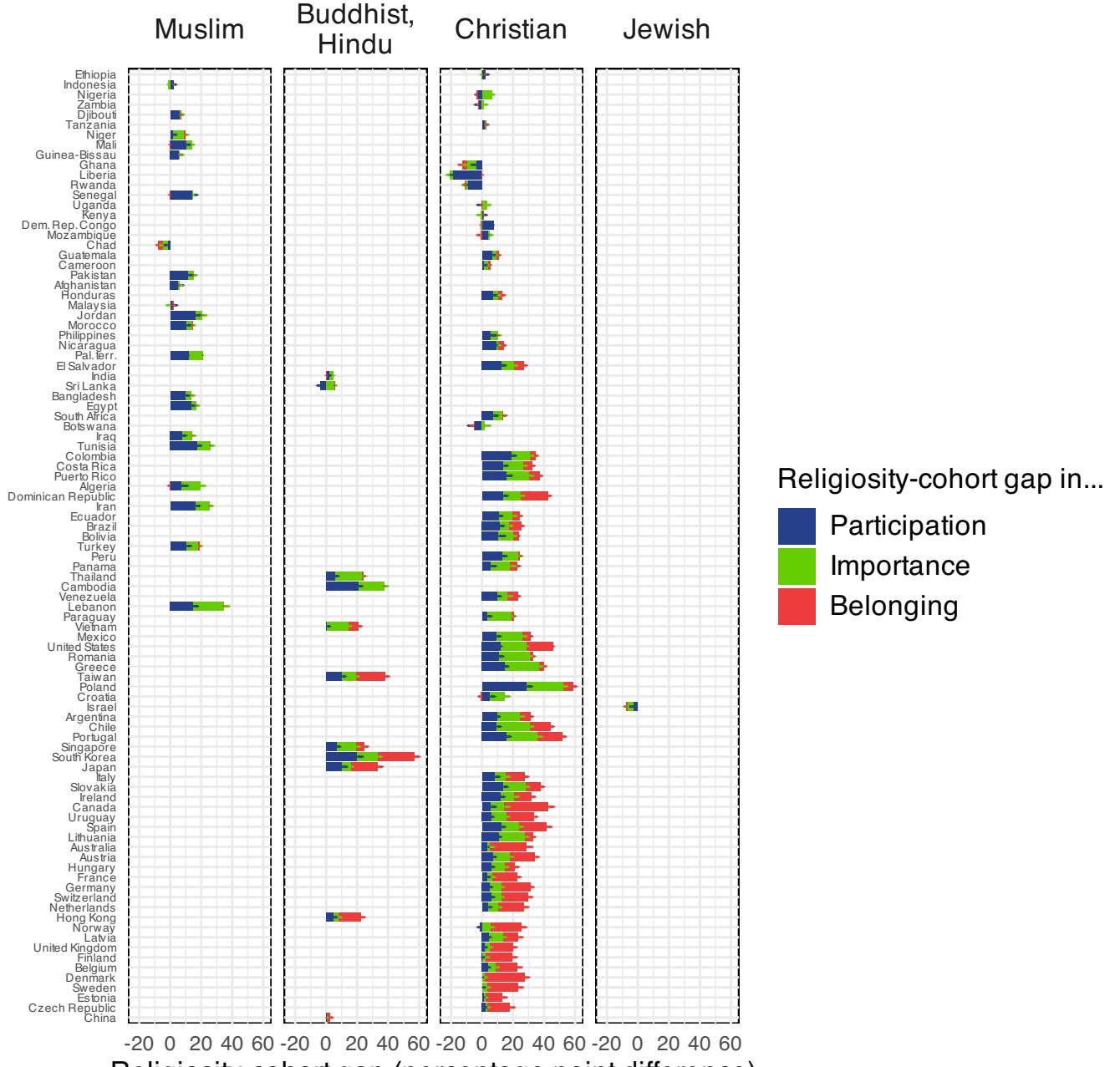

**Fig. 4 | Religiosity-cohort gap in participation, importance, and belonging in different countries and majority religions (country secularity ranked).** Pew data. All countries shown, except eastern post-communist countries. Differences in percentage of weekly attendance, high importance of religion and religious belonging between younger (< 40) and older (+ 40) individuals in different countries. Positive differences show younger individuals being less religious than older individuals. The countries on the y axis are ranked from most religious (top) to least religious (bottom). We plot the 95% confidence intervals. For better visibility, only half of the confidence interval is plotted - for positive percentage values at the right of the respective percentage bar, for negative percentage values at the left of the respective percentage bar.

Our post-hoc explanation for this finding is that eastern post-communist countries combine two things: on the one hand, their natural level of religiosity was artificially suppressed during the communist regime; on the other hand, they have lived through a nationalist-religious revival after the breakdown of the Soviet Union after 1990[25–27]. Hence, for now at least, eastern post-communist countries do not seem to follow the P-I-B sequence. While western post-communist countries have also seen an artificial repression of religion during the communist regime, they have not lived through an important nationalist-religious revival after 1990[28,29]. To allow for this specificity, in our overall model, we use a dummy variable for Eastern post-communist countries.

**Replicating the P-I-B sequence cross-sectionally**

To test the generalizability of our finding, we replicated the analysis with wave 7 of the World/European Value Survey (WVS/EVS 7). The replication generally confirms the findings with the Pew data: Again, we find one dimension of religiosity. A principal component analysis finds that the first component explains 85% of the variance. Again, the indicators for participation, importance, and belonging decline in a sequence, and again the cohort differences between older and younger rise and fall first for participation, then importance, and then belonging.

We again find that all differences are significantly different from zero. For younger cohorts, we observe: participation – importance:

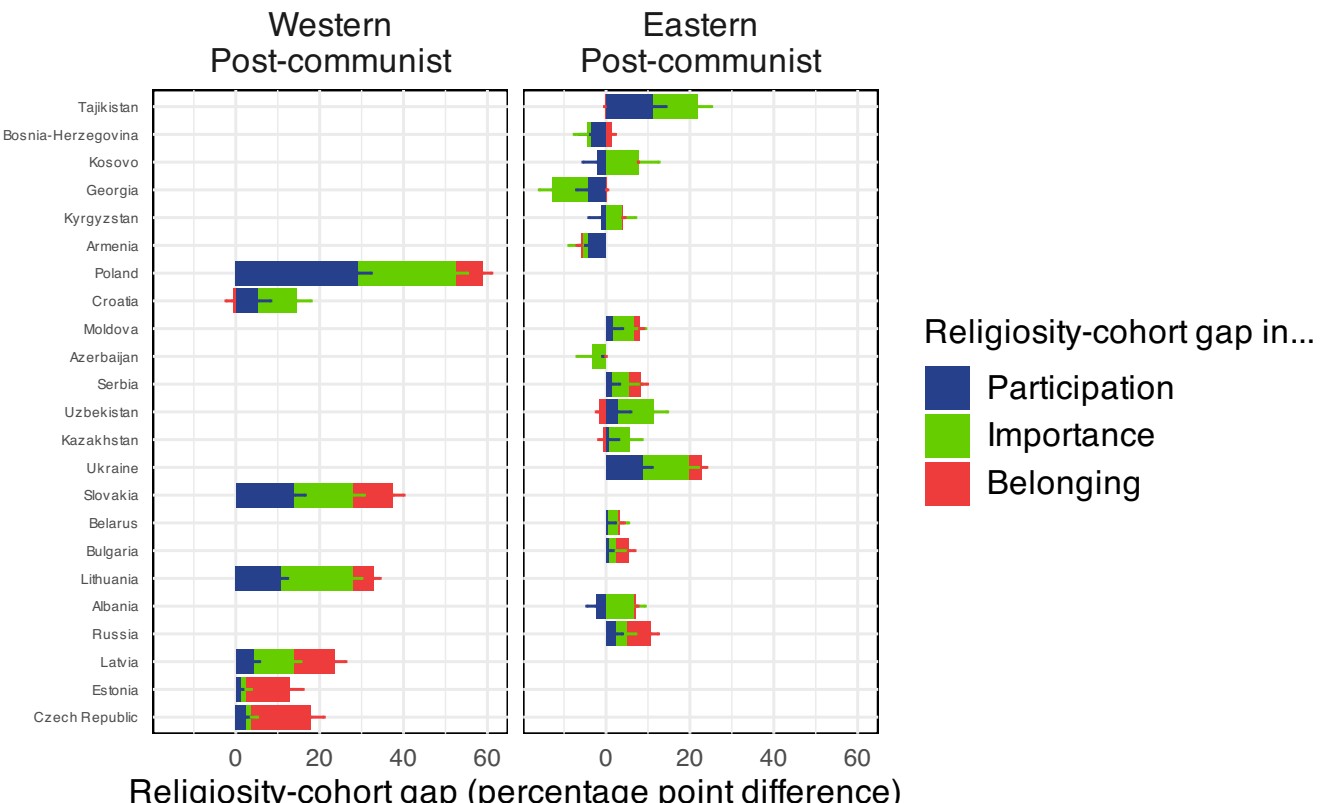

**Fig. 5 | Religiosity-cohort gap in participation, importance, and belonging in post-communist countries (country-secularity-ranked).** Pew data. Differences in percentage of weekly attendance, high importance of religion and religious belonging between younger (< 40) and older (+ 40) individuals in different countries. Positive differences show younger individuals being less religious than older individuals. The countries on the y axis are ranked from most religious (top) to least religious (bottom). We plot the 95% confidence intervals. For better visibility, only half of the confidence interval is plotted - for positive percentage values at the right of the respective percentage bar, for negative percentage values at the left of the respective percentage bar.

−0.623 [−0.763, −0.503]; participation – belonging: −1.440 [−1.601, −1.296]; importance – belonging: −0.817 [−0.907, −0.732]. For older cohorts, the corresponding contrasts are similarly large and negative: participation – importance: −0.559 [−0.687, −0.443]; participation – belonging: −1.433 [−1.588, −1.292]; importance – belonging: −0.874 [−0.974, −0.784]. Additionally, we find significant generational differences between younger and older respondents in each individual indicator: participation: −0.201 [−0.359, −0.049]; importance: −0.137 [−0.186, −0.090]; belonging: −0.194 [−0.308, −0.084].

The replication with WVS/EVS 7 permits checking the whether the P-I-B sequence is robust to the control of the gender ratio and separately for women and men (Supplementary Methods 3 and 4).

Comparing Pew and WVS/EVS 7 (Fig. 6), we note remarkably similar results. In both cases, younger cohorts have lower estimates than older cohorts and participation has lower estimates than importance which in turn has lower estimates than belonging. The WVS/EVS 7 dataset shows generally somewhat lower values of the sigmoid midpoints, which is caused by the inclusion of a different mix of countries in the two datasets. Even so, the size of the differences between midpoints of participation, importance, and belonging is very similar; the P-I-B sequence is thus reproduced very clearly in our replication.

The combined results using the Pew and the WVS/EVS 7 data affirm our cross-sectional hypothesis H1: Very religious countries show mainly cohort differences regarding public ritual attendance; countries with intermediate religiosity show cohort differences regarding all three indicators; countries with very low religiosity show cohort differences mainly regarding nominal belonging.

## Replicating the P-I-B sequence longitudinally

The following analysis investigates whether the proposed sequence can also be found in a longitudinal perspective. If our model is correct, we will find that countries at the beginning of the secular transition start with declining attendance but keep importance and belonging at high levels; countries at intermediate levels of the secular transition should show decline in all three dimensions; countries towards the end of the transition should have very low participation and importance rates and be characterized mainly by declining belonging rates. This is our longitudinal hypothesis H2.

The main difficulty of such an analysis lies in the fact that in the best cases, we can observe countries during a time period of 40 years, while the probable length of the secular transition is, according to Voas' theory of the secular transition, more than 200 years[1,2]. We tackle this difficulty by assuming a constant speed of the secular transition across countries. Furthermore, we assume that countries find themselves at different points of the secular transition, in what we call theoretical time (see Supplementary Methods 14). It then becomes possible to estimate a scaling factor that indicates just how fast the purported process runs and, accordingly, how long the secular transition usually lasts.

In the WVS/EVS within-country analysis, we include all countries that were surveyed at least 5 times since the beginning of the WVS/EVS enterprise. Our remaining sample consists of 17 geographically diverse countries, including Australia, Nigeria, South Africa, Mexico, the United States, Turkey, Sweden, Spain, India, and China. We exclude Russia, since it is an eastern post-communist country and thus shows a different trajectory. We run a non-linear regression model with the

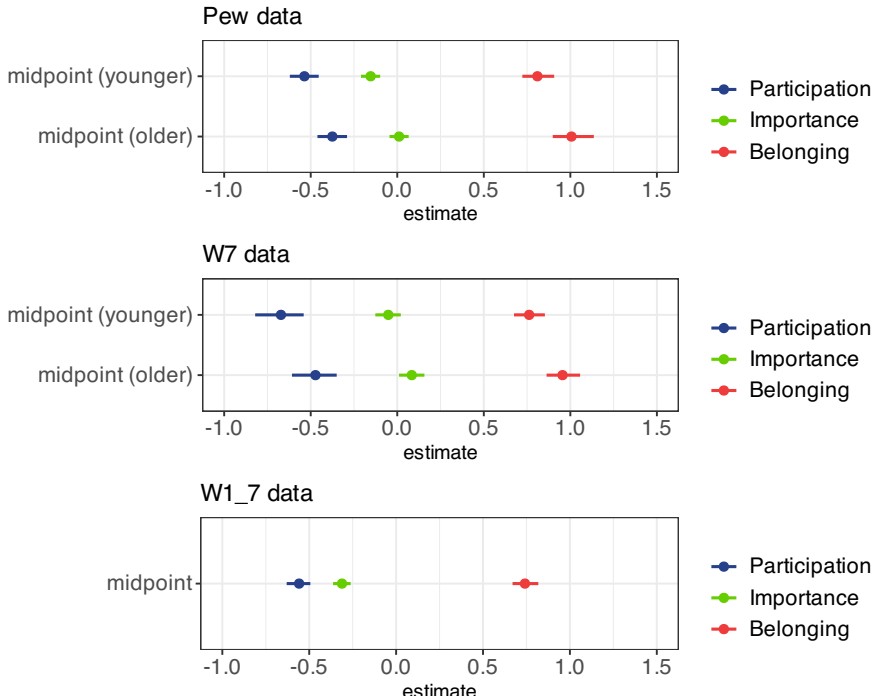

**Fig. 6 | Coefficients of midpoints in the cross-sectional and longitudinal models for participation, importance of religion, and belonging.** Based on all countries. With Pew data and WVS/EVS 7 data, the cross-sectional cases, we visualize the midpoints for older (40 + ) and younger ( < 40) individuals for the sigmoid models predicting percentage of weekly participation, high importance of religion, and religious belonging. With WVS/EVS 1-7 data, the longitudinal case, we visualize the sigmoid models predicting the three variables without distinguishing between older and younger cohorts. The lines represent highest (posterior) density intervals at the 95% level. Since the models are not linear, the intervals do not have to be symmetric. We control for continent, government support, government regulation, government discrimination, and pct migrants. In the longitudinal case, we omit the control for continent.

following specification:

$$RI = \left(1 + e^{a\left(CS\lambda^{-1} - b - BX\right)}\right)^{-1} \qquad (2)$$

where:
RI = religiosity indicators
CS = country secularity
$a$ = steepness of the curve
$b$ = intercept for the midpoint of the curve
$\lambda$ = scaling constant that transforms our initial relative time to theoretical time
$CS\lambda^{-1}$ = theoretical time
B = vector of coefficients for control variables
$X$ = vector of control variables

In this way, we seek to put the different countries onto a continuum of theoretical time representing the secular transition. Countries with higher aggregate religiosity are supposed to be earlier on this continuum. Since the shape of decline of the religiosity variables is supposed to be a sigmoid function, we can find a factor lambda that maximizes the fit of the country slopes to the supposed sigmoid functions. This factor lambda will also then give us information on the supposed length of the secular transition. The solution found is visualized in Fig. 7. The overall trends are observable, and some countries show an almost perfect fit (e.g., Argentina, Australia). Nevertheless, the fit is far from perfect. For example, South Africa is showing religious revival instead of decline on all indicators. The fit for changes in belonging is least satisfactory.

The within-country model shows, first a decline of participation, then a decline in importance, and lastly a decline in belonging, just as in the previous between-country models. The longitudinal perspective thus confirms the overall model of the three stages of the secular transition. Of course, the model is much less clear about what happens in the tails of the distributions. Participation and importance will likely not start out at 100% and all three indicators will probably not drop to 0%, but settle on a floor level of a few percent[30]. An additional result of this analysis is the insight that the secular transition seems to have the length of close to 250 years, and thus to take somewhat longer than suggested in previous literature on the basis of an analysis of European countries[2].

The results of our longitudinal analysis offer support for the main hypothesis but should be approached with caution. First, the model does not adequately fit the data for some countries. Second, the findings rest on the assumption that the average pace of the secular transition remains constant—an assumption that may not hold true. Lastly, the analysis assumes that data from a forty-year period can be extrapolated to predict trends over more than two centuries, including the assumption that, once begun, the secular transition would remain unaffected by major disruptions such as shifts in state policy, nationalistic conflict, or independence movements.

That having been said, we analyze the richest dataset currently available, offering the best existing test for our purpose. Significance tests, based on 95% highest posterior density intervals, show that differences among the three core indicators—participation, importance, and belonging—are consistently large and statistically robust across all waves of the WVS/EVS. Specifically, for the full pooled dataset (WVS/EVS Waves 1–7), we find: Participation – Importance: –0.280 [–0.356, –0.203]; Participation – Belonging: –1.297 [–1.385, –1.213]; Importance – Belonging: –1.017 [–1.104, –0.933].

Comparing the cross-sectional and longitudinal models, we note that the effect sizes are rather similar. In the longitudinal model we again see that the midpoints differ significantly in the expected

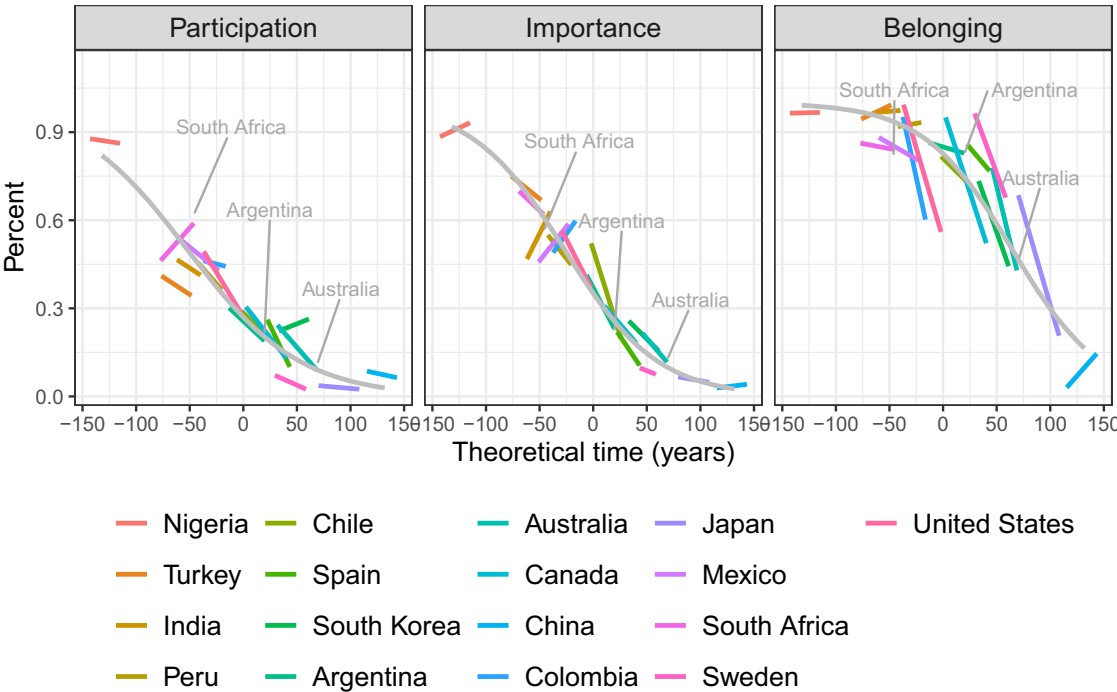

**Fig. 7 | Development of participation, importance of religion, and belonging in theoretical time.** WVS/EVS 1-7 data. We show 17 countries that were surveyed 5 times or more often, except eastern post-communist countries. Colored lines show the development of percentage of weekly participation, high importance of religion and religious belonging over time modeled with linear models. Country positions are found by shifting the countries towards the left or right according to their overall religiosity, multiplied by an expansion factor lambda.

manner. The coefficient for importance is closer to the coefficient of participation than in the Pew data and the WVS/EVS 7 data. This may again be due to the specific selection of countries used in the WVS1-7 dataset.

## Discussion

We have presented a revised model of secular transition according to which cohorts drop dimensions of religiosity in a specific order over time, abandoning higher involvement dimensions sooner than lower involvement dimensions. According to the model, religiosity differences show up first in public ritual attendance, then private importance of religiosity, and finally nominal belonging (the P-I-B sequence). The model implies rationality in the sense of dropping more costly religious traits first (Supplementary Discussion 1).

This model addresses the puzzle of variation in cohort differences across countries and indicators of religiosity worldwide. It predicts that a country at a certain threshold of modernization will enter a secular transition process. During this process, younger cohHorts become consecutively less religious than older cohorts.

The P-I-B sequence can be perceived in traditionally Muslim, Buddhist and Hindu, as well as traditionally Christian countries. In traditionally Christian countries, the full sequence can be observed. In traditionally Muslim countries, only the two first steps become apparent. In traditionally Buddhist/Hindu countries, the sequence is clearly discernible, but the estimations for participation are to be interpreted with caution since participation was measured differently in different surveys. The P-I-B sequence can furthermore be demonstrated both for women and men.

One group of countries does not follow the predictions of the model: eastern, post-communist countries. We argue that the specific history of a communist past with state repression and a societal crisis in 1990, has led to a transitory nationalist-religious upswing that currently masks the longer secular transition trend.

Our results challenge skeptics who doubt that secularization theory can be broadly generalized. Skeptics suggest the secularization observed in some regions, such as countries in Western Europe, is not comparable to dynamics elsewhere in the world[31–34]. According to skeptics, no general mechanism of secularization can be discerned. Instead, they suggest, we can at best discuss multiple secularities[35,36]. Modernization, secularization, and resurgence processes are expected to be contingent on highly complex and specific geographic, cultural, and historical conditions[36]. While we do not doubt that historical contingencies are important, our findings suggest that secularization processes may be more similar across the world than was previously thought.

It is important to note, however, that our findings do not imply that the entire world is becoming less religious in the short term. As previous research has demonstrated, the opposite may be true, since countries with a low score on the human development index (HDI) are more religious, have higher fertility rates, and have populations growing at above-average speeds[17]. As a result, in the short to medium term, religiosity could increase globally because more religious countries make up a growing share of the world's population. However, if the secular transition outlined here continues—the world will see a decline in religious belief and practice in the long run.

Our study has some limits. First, our data do not permit us to draw firm conclusions with respect to traditionally Muslim countries. We expect that they will continue to follow the secular transition, eventually including the third step of the transition, but only time will tell whether this prediction will turn out correct. Second, our analysis takes countries as basic units - all internal variation in the countries, therefore, remains hidden. Third, we acknowledge exceptions to our model, such as eastern post-communist countries, that do not confirm the predictions. Fourth, we do not go into questions of how spirituality develops in these countries. However, in our view, the rise of what is called spirituality in the West is equally a form of secularization and

represents an intermediate state between strong religiosity and secularity[37]. A final and arguably the most important point is the limited temporal scope of the longitudinal data, which warrants caution. We are assuming that these 40 years, across different countries, reflect distinct stages of a shared trajectory spanning over 200 years. This is an important assumption that could ultimately prove inaccurate.

These limitations are offset by important strengths. Previous accounts had theorized the secular transition as uniform. We propose that the secular transition follows three distinct steps. Since modernization renders religious solutions to life's problems less efficient, populations drop religious traits in the P-I-B sequence. Previous tests of the secular transition model were limited to the western world. Our analysis applies and tests the model successfully across all continents. Previous studies suggested that the secular transition model might be restricted to traditionally Christian countries. Our analysis reveals the model may apply just as well to countries that are traditionally Muslim, Buddhist or Hindu (more time will need to pass before we know conclusively). An additional strength of our study lies in the robustness of our findings across multiple datasets. Specifically, we observe consistent evidence for the hypothesized three-step pattern in both the Pew dataset and in analyses of the WVS/EVS data, the latter replicated using both cross-sectional and longitudinal designs.

How do religious resurgences fit with this model of the stages of the secular transition? The 20th century has witnessed notable religious resurgences, including the emergence of the Pentecostal movement, the widespread resurgence of Islam in various countries since the 1970s, and the Orthodox resurgence in the aftermath of the Soviet Union's disintegration[38]. Our answer is multifaceted. The combined influence of communist repression, the disintegration of the Soviet Union, and the Orthodox resurgence is evident in the data. Countries that have experienced these three phenomena do not conform to the expected pattern predicted by the model, instead exhibiting limited or no cohort differences in religiosity indicators. It is possible that the interplay of secularization and resurgence cancel each other out, but further investigation is necessary to confirm this. However, the Pentecostal and Muslim resurgences do not appear in our data. Our hypothesis is that these resurgences are an inherent aspect of the broader process of modernization and secularization. While they temporarily reinvigorate religion within society, they do not alter the overarching secularization trend.

The results of our study also raise a question regarding the apparent absence of importance of state influence. Is it possible that there exists a broad secular transition trend, despite varying religious policies across different states? It is well-established that state policies can have a marked impact on the presence, magnitude, and pace of secular transition[14]. However, their influence is often limited in the long term. As a case in point, South Korea experienced a resurgence of Protestant and Buddhist religions due to state regulation in the late 1960s and 1970s, yet this resurgence was transitory in nature[39].

Our study has demonstrated that increasing country secularity coincides with the P-I-B sequence. By doing so, we have sidestepped the question of just what non-religious factors causally explain country secularity in the first place. What necessary and sufficient conditions may lead a country to enter the secular transition and to keep it moving along the transition trajectory? While it is obvious that indicators of modernization, such as HDI, are linked to the level of country secularity[16,18,40] and to the religiosity-cohort gaps on an aggregate level (Supplementary Discussion 2), the underlying causal mechanisms have yet to be disentangled.

An interesting implication of our model is that we can expect individual legitimations and rationalizations of the process that can be observed both in quantitative, but especially qualitative research. As individuals abandon previously socially expected traits, they may encounter a perceived social disapproval. To mitigate this perception, they may rationalize that their social worth remains intact despite dropping the trait. During the P-I-B sequence, individuals may argue that they can still find religion important and be a "good religionist" even after giving up participation. Similarly, even when having stopped considering religion important in their life, they may argue that they still belong to the religious group by tradition. And even after giving up religious belonging, they may argue that they can still maintain "good values."

Summing up, our results suggest a worldwide secular transition has been underway for many decades. Religion seems to be declining in three stages worldwide.

## Methods

### Data

We use three data sets for this paper: An integrated Pew Research Center data set, wave 7 of WVS/EVS and a subset of the integrated WVS/EVS dataset with countries that were surveyed more than 5 times or more often (17 countries).

In Table 1, we give some descriptive information on the number of countries, mean number of time points per country, and mean and standard deviation for every indicator in the Pew and WVS/EVS data.

The Pew survey data includes cross-sectional data points from more than 100 countries described in a 2018 Pew report.[18] The data were compiled from survey projects including the Sub-Saharan Africa Survey, the Global Survey of Islam, the Latin America Survey, the Asian Americans Survey, the Jewish Americans Survey, the U.S. Religious

**Table 1 | Descriptive information**

| Concepts | Indicators | Countries | Time points per country (mean) | Percentage (sd) |
|---|---|---|---|---|
| **Pew** | | | | |
| Participation | Weekly attendance ore more | 111 | 1 | 39.8 (27.0) |
| Importance | High importance of religion | 111 | 1 | 53.8 (31.2) |
| Belonging | Identify with a religion | 111 | 1 | 87.9 (17.1) |
| **WVS/EVS 7** | | | | |
| Participation | Weekly attendance ore more | 58 | 1 | 32.8 (21.8) |
| Importance | High importance of religion | 58 | 1 | 49.9 (31.6) |
| Belonging | Identify with a religion | 58 | 1 | 78.7 (24.9) |
| **WVS/EVS 1-7** | | | | |
| Participation | Weekly attendance ore more | 17 | 5.76 | 31.4 (20.8) |
| Importance | High importance of religion | 17 | 5.76 | 38.4 (25.4) |
| Belonging | Identify with a religion | 17 | 5.76 | 75.4 (23.5) |

Based on all countries. Pew surveys: 2008-2023; WVS/EVS 7: 2017-2022, WVS/EVS 1-7: 1981-2022.

Landscape Study, the Central and Eastern Europe Survey, the Western Europe Survey, the Muslim Americans Survey, and Pew Global Attitudes surveys. Because Burkina Faso measures are incomplete, we omit this country. Additionally, we add data from more recent Pew religion survey projects conducted in India, South/Southeast Asia and East Asia. Altogether, we use Pew data from 111 countries surveyed between 2008 and 2023.

The WVS/EVS survey data is an integrated dataset including seven waves, and both WVS and EVS data[41]. Wave 7 of WVS/EVS is used as a replication of the analysis of Pew data. Every included country is surveyed only once. The dataset gives information on 58 countries. The measures are very similar to those in Pew.

The third dataset consists of countries that were surveyed more than 4 times in WVS/EVS between 1981 and 2022. The 17 countries in question are: Argentina, Australia, Canada, Chile, China, Colombia, India, Japan, Mexico, Nigeria, Peru, South Africa, South Korea, Spain, Sweden, Turkey, United States.

### Analytic strategy

We model the indicators for participation, importance, and belonging as sigmoid functions of country secularity both for older (40 + ) and younger individuals (40−). The three outcomes are estimated jointly within a system of three equations that together define the model likelihood.

The central independent variable "country secularity" is simply the first principal component of a principal component analysis of the three standardized variables attendance, importance of a religion, and identifying with a religion.

We then test (1) Whether the midpoints of the sigmoid functions are different for older and younger cohorts and for the three different indicators in a cross-sectional perspective as predicted by Hypothesis H1 (older cohorts should be more religious than younger cohorts; participation should drop before importance and importance before belonging); and (2) whether the midpoints of the sigmoid functions are different for the three different indicators in a longitudinal perspective as predicted by our Hypothesis H2 (participation should drop before importance of religion and importance of religion before belonging). We run different models with our preferred model adjusting for continent, government support, government regulation, government discrimination, and migration.

It is not useful to adjust for majority religion or cultural zone since the categories of these two variables do not contain enough variation of the dependent variable (a problem of "unbalanced design" of our datasets). While this problem already appears for continent, it is aggravated for majority religion and cultural zone. To test H1, we first use the cross-sectional Pew data. These findings are replicated with the cross-sectional WVS/EVS data (wave 7). To test H2, we use a subsample of the WVS/EVS data (waves 1-7) containing the 17 countries with more than 4 measurements. Here, we find a factor lambda by which countries with lower country-secularity are "pushed back" in "theoretical time". This value gives us information on the supposed length of the secular transition and permits calculating whether the midpoints of the sigmoid functions are different for the three different indicators as predicted by H2.

We acknowledge that our analysis does not explicitly incorporate uncertainty in the scores of our dependent variables participation, importance, and belonging, as well as our independent variable, country secularity. While we considered addressing this issue using latent structural equation modeling, we ultimately opted against it to maintain the tractability of an already complex model. However, robustness checks (Supplementary Methods 7,8,10,11) indicate that our findings are consistent across different ways of measuring and specifying our dependent and independent variables.

In this paper, we combine a frequentist and Bayesian approach. The confidence intervals in Figs. 1,3,4, and 5 follow a frequentist logic.

All other inferences use a Bayesian logic. We follow Gelman and Shalizi in their view that both frequentist and Bayesian methods can be combined pragmatically in a hypothetico-deductive scientific framework[42,43]. We mainly use Bayesian methods when the relatively complex functions needed a flexible estimation approach.

### Dependent variables

The dependent variable in the single cross-sectional analyses is the percentage difference of religiosity between "older" (40 + ) and "younger" (under 40) individuals. We calculate this difference separately with the dichotomous religiosity indicators weekly ritual attendance (or not), seeing religion as very important in one's life (or not), and identifying with a religion (or not). Distinguishing only two age-cohorts - "younger" and "older" - permits illustrating broad trends in a simple way. We conduct robustness checks to show that different thresholds do not change results substantively.

*Religious participation* was measured, in the Pew surveys, with slightly different wordings for different religions. The general formulation for non-Muslim populations was "Aside from weddings and funerals how often do you attend religious services… more than once a week, once a week, once or twice a month, a few times a year, seldom, or never?" In the East Asia survey, which included Vietnam (a country not usually considered part of this geographic region), the most prevalent religions, including Buddhism, do not emphasize weekly communal worship to the degree that is common in Abrahamic religions. Respondents in these countries were asked separate questions about whether they generally go to a shrine, temple, church, or monastery. If respondents in East Asia said "yes" to at least one of these measures, we classify them as regular participants in religious rituals. For Jews, the beginning of the question was phrased as: "Aside from special occasions like weddings, funerals and bar mitzvahs, how often do you attend Jewish religious services at a synagogue, temple, minyan or Havurah". For Muslims, one of the slightly differing wordings was: "On average, how often do you attend the mosque for salah? More than once a week, once a week for Friday afternoon Prayer, once or twice a month, a few times a year, especially for Eid, seldom or never?". In the WVS/EVS, religious attendance was measured with the question "How often do you attend religious services?". The response options were "more than once a week", "once a week", "once a month", "only on special holy days", "once a year", "less often", "never/practically never".

*Importance of religion* was measured, in the Pew surveys, with the question: "How important is religion in your life – very important, somewhat important, not too important, or not at all important?". In the WVS/EVS, importance of religion is measures with the question: "For each of the following, indicate how important it is in your life. Would you say it is: very important, rather important, not very important, not at all important." The respondents reacted to the Item: "Religion".

*Belonging* is measured as a dichotomous variable indicating whether the individual claims to belong to a religion or not. The Pew question is: "What is your present religion, if any?" Respondents are given a country-specific list of response options, including several major world religions, as well as "atheist," "agnostic" or "nothing in particular". In the WVS/EVS, the question was asked: "Do you belong to a religion or religious denomination? If yes, which one?"

Both for Pew and WVS/EVS surveys, we transform all indicators of religiosity into dichotomous variables with 1 = weekly attendance, and 0 otherwise; 1 = seeing religion as very important in one's life, and 0 otherwise; and 1 = being affiliated with a religion, and 0 otherwise. We then calculate the percentage of individuals per country with the respective religiosity indicator.

The dependent variables in the WVS/EVS repeated cross-sectional analyses are the percentages of individuals with weekly attendance (or more), who find religion very important in their lives, and who claim to be affiliated to a religion - all of these per country and year. We conduct

robustness tests to show that different cut-offs do not lead to different results (Supplementary Methods 11).

## Independent variables

Our main independent variable is the position of a country on an assumed secular transition trajectory in a given year. Using multidimensional scaling techniques, we constructed a latent dimension of the variables participation, importance of religion, and belonging[44]. It turns out that a solution with two dimensions and interval scaling fits the data very well. We keep the first dimension which contains the main information of country secularity. This procedure leads to a substantively similar solution as when keeping the first principal component of a Principal Component Analysis (PCA).

In the longitudinal WVS/EVS analysis, our second independent variable is survey year.

## Controls

We test whether our findings hold up when controlling for a number of control variables.

We briefly justify our controls by specifying how they might be thought to interfere with country secularity and the P-I-B sequence.

**Eastern post-communist.** This dummy variable is set to 1 for the countries "Belarus", "Bulgaria", "Georgia", "Moldova", "Montenegro", "North Macedonia", "Russia", "Serbia", "Ukraine", "Armenia", "Kazakhstan", "Kyrgyzstan", "Uzbekistan", "Tajikistan", "Uzbekistan", "Bosnia-Herzegovina", "Kosovo", "Azerbaijan", "Albania" and to 0 otherwise. While we initially used a dummy variable for post-communist countries in general, it turned out that a dummy variable distinguishing only eastern post-communist countries gives a better fit.

**Continents.** We distinguish the continents Africa, Americas, Europe, and we combine the levels Asia & Oceania. Continents might influence country secularity and the P-I-B sequence since cultural diffusion is easier within than across continents. Continents may also have specific mixes of economic development and dominant religions.

**Historical religion.** This variable assigns the largest religion found in a given country at the assumed start of the secular transition. For example, it assigns Christianity to Estonia (even though a large share of Estonians do not report identifying with a religion), Islam to Pakistan, and Hinduism to India. We coded this variable using Pew Research Center's estimates of religious composition[45]. Historical religion of the country might influence both country secularity and the P-I-B sequence since different religious cultures may put different emphasis on different dimensions of religion.

**Cultural zones.** Following Haerpfer[41], we distinguish the following nine cultural zones: English-speaking, Latin America, Catholic Europe, Protestant Europe, African-Islamic, Baltic, South Asian, Orthodox and Confucian. We use operationalization of the Inglehart/Welzel[46] tradition since this typology has been empirically shown to be important. We do not use the variables historical religion and cultural zones in the same models because of too much overlap.

*Government support, government regulation,* and *government discrimination* are measured with the respective variables from the RAS dataset[47]. Government support measures "different ways a government can support religion including financial support, policies which enforce religious laws, and other forms of entanglement between government and religion"[48]. Government regulation measures "different ways governments regulate, restrict, or control all religions in the state including the majority religion"[48]. Government discrimination measures "restrictions that are placed on the religious institutions and practices of religious minorities that are not placed on the majority group"[48]. We have dichotomized the three continuous variables,

distinguishing a high (1) and a low value (0). Government support of religion may suppress country secularity, while government regulation and discrimination may boost it[49–52].

**Gender.** We use both the (self-reported) dichotomous variable gender (female = 1, male = 0), and the gender ratio expressed as the percentage of women in the respective population or subpopulation. A large literature shows that women are more religious than men in western countries, even when controlling for various socio-structural variables[50,53,54]. A world-wide assessment, however, suggests that such a gender gap is mainly present in traditionally Christian countries[19,55]. In our models, we use gender ratio as a control variable. See Supplementary Methods 3 where it is shown that the P-I-B is robust to including or excluding the control variable gender ratio (and other control variables). See furthermore Supplementary Methods 4, where the P-I-B sequence is calculated separately for women and men.

**Percentage of migrants.** This is the migrant stock as a percentage of the total population in 2017[56]. Since migration will often happen from less to more economically developed countries, and since religiosity is more prevalent in less developed countries, higher migrant stocks may depress country secularity. We dichotomize the variable into a migrant stock of less or equal and a migrant stock of more than 10%.

We do not control for cultural zones and traditional religions in our final models since the categories of these variables do not contain enough variation of the dependent variable. We have also constructed a variable measuring presence/absence of wars, but do not use it in our analyses since it correlates too strongly with the secular transition variable. Further analyses concerning the variable cultural zone can be found in Supplementary Methods 5.

We use weighted data for the Pew and the WVS/EVS survey. In the aggregated Pew dataset, there were no missing values in dependent variables. Missing values in independent and control variables were less than 1%. In the WVS/EVS 7 data, missing values in dependent variables was smaller than 3%. The highest percentages of missing values were in variables used for robustness tests such as prayer (6.1%) or belief in God (6.8%). In the Pew and WVS/EVS 7 datasets we imputed missing values with predictive mean matching (method = "pmm") using the mice library (multivariate imputation by chained equations). We used 10 imputed dataset (m = 10). In the WVS1-7, we imputed missing values of the variables participation, belonging, and confidence in the church (all three with 2% of missing values) with a LOCF/NOCB (forward/backward fill) imputation procedure inside the countries and sorting for year. We imputed missing values in the variable importance of religion (with 9.8% of missing values) with a two-level Bayesian linear mixed model approach (mice package, method = "21pan", predictors: confidence in church, participation, belonging). The control variables government support, government regulation, and government discrimination had 21.6% of missing variables. These values were imputed with LOCF/NOCB, since we did not have evident predictors for a "21pan" approach. In a robustness test (Supplementary Methods 12), we show that imputing or not imputing missing values does not change the results substantively. The data analysis was conducted in R, version 4.1.1.

## Outliers

We calculated different measures of fit for every country, across indicators and for older and younger cohorts. We focus specifically on the deviation delta for every country, that is, a deviation measure for the difference between the predicted and the real difference between the older and the younger cohort in that country across indicators (Supplementary Methods 6). Two groups of countries show a relatively bad fit with respect to this measure in the Pew data: In some post-communist countries such as Albania, Azerbaijan, or Georgia, younger people are actually more religious than older people, whereas in some

Asian countries such as Vietnam, South Korea, Taiwan, or Hong Kong, younger people are less religious than older people - but too much so (with respect to the model). The United States shows also a rather high deviation from the model (with younger people being too secular with respect to older people).

## Validity issues and robustness checks

In what follows, we describe how we have addressed possible validity issues.

(1) In our analyses, we have assumed that the cohort differences with younger cohorts being less religious than older cohorts are caused by cohort effects and not life-cycle effects. This assumption may be criticized. We acknowledge that the evidence corroborating the assumption is stronger for western than for non-western countries. However, proponents of a life-cycle argument would have to explain why - if their assumptions were correct - in very religious countries, people would have more participation as they age, in medium religious countries, people would have more participation, importance, and belonging as they get older, and in very secular countries, the population would increase affiliation as it ages. It seems unlikely that someone would try to make such an argument.

(2) Another issue for our analysis concerns measurement invariance[57]. Indicators of religiosity may be non-comparable across countries since the actions, beliefs and values measured mean something different in different religions. In congregational religions such as Christianity or Islam religionists gather for public rituals in a community or congregation and often see their identity strongly in terms of this community. In non-congregational religions such as Shinto, Buddhism or Hinduism, the emphasis lies more on individual practice and public rituals are seen as beneficial, but not necessary[23]. Public rituals are not held in a regularly occurring religious service of a group (or congregation); rather, individuals and families seek out temples and shrines when they feel the need to do so. There is often no felt belonging to an imagined large confessional group, but there may be the strong feeling of belonging to an ethnic or caste group; furthermore, individuals and especially families have links to specific temples or shrines; however, they may also worship at other places and to gods, buddhas, or kamis of different religious traditions.[58-61] The effect may then be that, assuming a real latent level of religiosity, the indicator over- or underestimates religiosity.

We address the problem by analyzing cohort differences, comparing an older and a younger cohort in every country. This controls for possible bias. The problem remains, however, when we construct our measure of country secularity where biased estimators may bias the positioning of the countries. To address this point, we conduct robustness tests with different ways of constructing country secularity (Supplementary Methods 7). On the subject of measurement invariance, our model makes an additional point. Studies that test measurement invariance for religiosity items over a large number of countries and that assume all countries to be "at the same time" may get it wrong. They may conclude that there is measurement invariance, while, in fact, countries are at different points in time on the secular transition and therefore *must* show different correlational structures.

(3) An important question is the choice of indicators of religiosity. What we measure with "importance of religion in one's life" is an indicator for personal religiosity in a very broad sense including religious beliefs and personal practices such as prayer. These different elements, and especially belief variables vary strongly across different religious and cultural areas. We have therefore chosen to use only one indicator both in the Pew study and the WVS/EVS data, whose meaning seems to be relatively well

comparable across cultures and who clearly taps personal religiosity. As a robustness check, we investigate how well our results hold when measuring the "importance" dimension in two other ways: (a) with a composite measure of importance of religion and prayer; and (b) with a composite measure of importance of religion, importance of god, belief in god, and self-description as a religious person (Supplementary Methods 8).

(4) We do not use a measure of modernization (e.g., the HDI index) as an independent variable, since we do not seek to causally explain secularization. Neither do we use such an indicator as a control, since we do not expect the secular transition to happen independently of modernization. Controlling for modernization (e.g., with an indicator such as the HDI) would then introduce bias rather than reduce it.

(5) We model our three dependent variables using a sigmoid (inverse logit) function. We chose the logit function due to its intuitive interpretability compared to alternative link functions. However, one might question whether the results would differ under alternative specifications. A good alternative candidate would be the probit function (non-parametric functions are not feasible because of the relatively small number of countries). In Supplementary Methods 9 we show graphically that results look very similar with probit. AIC measures of logit and probit are also very close.

(6) Our results are furthermore robust with respect to (a) varying the cut point with respect to cohorts; (b) varying the cut points with respect to attendance and importance of religion; (c) including or excluding the controls; including or excluding imputed values; including or excluding the weights in WVS/EVS data. (Supplementary Methods 10,11,12).

## Midpoints of participation, importance, and belonging functions

In Table 2, we give the numbers visualized in Fig. 6 (above). These are estimates for midpoints of sigmoid functions of religiosity indicators of older and younger cohorts (Pew and WVS/EVS 7 data) and all cohorts (WVS/EVS 1-7).

## Credibility intervals for differences between midpoints of participation, importance, and belonging functions (within and between indicators)

We test whether the difference between the midpoints for older and younger cohorts within indicators and the difference between the

**Table 2 | Midpoints of sigmoid functions of religiosity indicators of older and younger cohorts (Pew and WVS/EVS 7 data) and all cohorts (WVS/EVS 1-7)**

|  | Participation | Importance | Belonging |
|---|---|---|---|
| **Pew between (2008–2023)** | | | |
| Midpoint older | −0.37 (0.04) | 0.01 (0.03) | 1.01 (0.06) |
| Midpoint younger | −0.53 (0.04) | −0.15 (0.03) | 0.81 (0.05) |
| **WVS/EVS 7 between (2017–2022)** | | | |
| Midpoint older | −0.48 (0.07) | 0.08 (0.04) | 0.96 (0.05) |
| Midpoint younger | −0.68 (0.07) | −0.06 (0.04) | 0.76 (0.05) |
| **WVS/EVS 1-7 within (1981–2022)** | | | |
| Midpoint | −0.56 (0.04) | −0.28 (0.03) | 0.74 (0.04) |

Based on all countries. These are the numbers visualized in Fig. 6 (above). Estimates for midpoints of sigmoid functions of religiosity indicators of older and younger cohorts (Pew and WVS/EVS 7 data) and all cohorts (WVS/EVS 1-7). Standard errors in parentheses. We show the values controlling for continent, government support, government regulation, government discrimination, and pct migrants. In WVS/EVS7 and WVS/EVS1-7, pct female is additionally controlled. In WVS/EVS 1-7 the control for continent is omitted.

**Table 3 | Credibility intervals for differences (within and between indicators)**

| Pew: | Difference | CI (95%) |
|---|---|---|
| Difference younger-older: participation | −0.162 | −0.253; −0.072 |
| Difference younger-older: importance | −0.164 | −0.203; −0.126 |
| Difference younger-older: belonging | −0.196 | −0.325; −0.080 |
| Difference younger: participation - importance | −0.383 | −0.453; −0.315 |
| Difference younger: participation - belonging | −1.346 | −1.455; −1.244 |
| Difference younger: importance - belonging | −0.962 | −1.050; −0.885 |
| Difference older: participation - importance | −0.386 | −0.460; −0.312 |
| Difference older: participation - belonging | −1.380 | −1.521; −1.258 |
| Difference older: importance - belonging | −0.994 | −1.117; −0.893 |
| **WVS/EVS 7:** | | |
| Difference younger - older: participation | −0.201 | −0.359; −0.048 |
| Difference younger - older: importance | −0.137 | −0.186; −0.090 |
| Difference younger - older: belonging | −0.194 | −0.308; −0.084 |
| Difference younger: participation - importance | −0.623 | −0.763; −0.503 |
| Difference younger: participation - belonging | −1.440 | −1.601; −1.296 |
| Difference younger: importance - belonging | −0.817 | −0.907; −0.732 |
| Difference older: participation - importance | −0.559 | −0.686; −0.443 |
| Difference older: participation - belonging | −1.433 | −1.588; −1.292 |
| Difference older: importance - belonging | −0.874 | −0.973; −0.784 |
| **WVS/EVS 1-7:** | | |
| Difference participation - importance | −0.280 | −0.356; −0.204 |
| Difference participation - belonging | −1.297 | −1.385; −1.213 |
| Difference importance - belonging | −1.017 | −1.104; −0.933 |

Based on all countries. Pew and WVS/EVS7 data. Significance tests using highest (posterior) density intervals, 95%. The first three rows test within the indicators whether younger cohorts are significantly less religious than older cohorts regarding participation, importance, and belonging. The next three rows show test whether the midpoints between indicators for the younger cohorts are significantly different. The last three rows do the same of the older cohorts.

midpoints between indicators (within older and younger cohorts) is significant. Because the distribution assumptions for *t*-tests were not met and because our models are rather complex, we use a Bayesian instead of a frequentist perspective to hypothesis testing as proposed by Bolstad[62] and Kruschke[63].

To do so, we calculate the highest (posterior) density interval at the 95% level for the difference in question[64]. To assess the evidence against the null hypothesis, we defined a Region of Practical Equivalence (ROPE) from −0.03 to +0.03 around the null value. If the 95% HPDI excluded this ROPE, we interpreted the result as evidence against the null hypothesis. This corresponds conceptually to a two-sided test in frequentist terms[63]. The prior distributions were chosen to be weakly informative. Priors for midpoints and slopes were assumed to be normally distributed with a mean of 0 and a standard deviation of 10, allowing values to vary widely. Priors for standard deviations were assumed to be t-distributed with 3 degrees of freedom, a location parameter of 0, and a scale parameter of 10. This distribution has heavier tails compared to the normal distribution, which makes it more robust to outliers. The scale parameter of 10 indicates that the standard deviation is expected to be relatively large. We employ a Hamiltonian Monte Carlo (HMC) algorithm with four chains and 5000 iterations, utilizing the No-U-Turn Sampler (NUTS) developed by Hoffman and Gelman[65]. The models converge. The fit diagnostics show satisfactory treedepth, satisfactory E-BFMI, and satisfactory effective sample size. There are no divergent transitions. We performed a local sensitivity analysis to evaluate the robustness of our findings to different prior specifications. Following Gill[66], we explored less informative priors. As further reducing the degrees of freedom was not advisable, we instead doubled the variance while keeping the degrees of freedom fixed and assessed the impact on the posterior distribution. The results indicate that this

adjustment has a negligible effect on the posterior parameters (Supplementary Methods 13).

It turns out that the differences between the midpoints of older and younger cohorts (within the indicators) and for either older or younger cohorts between the indicators in our overall analysis are all significant (Table 3). In other words, the credible intervals fall outside the region of practical equivalence (ROPE). This is true for the Pew data, the WVS/EVS 7, and the WVS/EVS 1-7 data.

This study is based exclusively on secondary analysis of publicly available, anonymized data from Pew Research Center and the World Values Survey (WVS/EVS). According to the University of Lausanne's research ethics guidelines, the use of such data does not require prior ethics approval. As the data are fully anonymized and collected with informed consent by the original organizations, this research qualifies as exempt from further ethical review.

### Reporting summary
Further information on research design is available in the Nature Portfolio Reporting Summary linked to this article.

## Data availability
The study uses a dataset compiled from multiple Pew Research Center surveys. The combined dataset is available in the replication package for this paper on the Open Science Framework (OSF) at [https://doi.org/10.17605/OSF.IO/VCZTA][67]. All underlying Pew datasets are freely available at https://www.pewresearch.org/datasets/. The surveys are: East Asian Societies Survey (2023) https://www.pewresearch.org/dataset/east-asian-societies-survey-dataset/ https://doi.org/10.58094/5jv2-m279. South and Southeast Asia Survey (2022) https://www.pewresearch.org/dataset/south-and-southeast-asia-survey-dataset/ https://doi.org/10.58094/rf31-hd47. India Survey (2019-2020) https://

www.pewresearch.org/dataset/india-survey-dataset/ https://doi.org/10.58094/RFTE-A185. Western Europe Survey (2017) https://www.pewresearch.org/dataset/western-europe-survey-dataset/ Survey of U.S. Muslims (2017) https://www.pewresearch.org/dataset/2017-survey-of-u-s-muslims/ Global Attitudes Spring 2017 Survey (2017) https://www.pewresearch.org/dataset/spring-2017-survey-data/ Eastern Europe Survey (2016) https://www.pewresearch.org/dataset/eastern-european-survey-dataset/ Global Attitudes Spring 2015 Survey (2015) https://www.pewresearch.org/dataset/spring-2015-survey-data/ Israel Survey (2015) https://www.pewresearch.org/dataset/pew-research-center-2015-israel-survey-dataset/ U.S. Religious Landscape Study (2014) https://www.pewresearch.org/dataset/pew-research-center-2014-u-s-religious-landscape-study/ Religion in Latin America Survey (2013-2014) https://www.pewresearch.org/dataset/religion-in-latin-america/ Global Attitudes Spring 2013 Survey (2013) https://www.pewresearch.org/dataset/spring-2013-survey-data/ Survey of Jewish Americans (2013) https://www.pewresearch.org/dataset/a-portrait-of-jewish-americans/ Asian Americans Survey (2012) https://www.pewresearch.org/dataset/asian-americans-2012/ The World's Muslims Survey (2010-2011) https://www.pewresearch.org/dataset/the-worlds-muslims/ Sub-Saharan Africa Survey (2008-2009) https://www.pewresearch.org/dataset/tolerance-and-tension-islam-and-christianity-in-sub-saharan-africa/ Use of the Pew data complies with the Pew Research Center's Terms and Conditions for data use. This study also uses publicly available data from the World Values Survey and European Values Study, available at https://www.worldvaluessurvey.org/wvs.jsp[41], accessed in accordance with the website's Terms of Use. An extract of the relevant WVS/EVS data that was used in this paper is provided on the Open Science Framework (OSF) at [https://doi.org/10.17605/OSF.IO/VCZTA][67].

## Code availability
The analysis code and replication materials used in this study are available on the Open Science Framework (OSF) at [https://doi.org/10.17605/OSF.IO/VCZTA]. The repository includes a README file with instructions for reproducing the results.

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

## Acknowledgements

We thank David Voas, Monika Wohlrab-Sahr, Jeremy Senn, and Rafael Lalive for discussions and comments on the manuscript. We also thank Simon G. Brauer for a code check. The project started when J.S. was a visitor at Nuffield College, invited by ND.DG. in 2023. C.H.'s work on this project was supported by Grant #63095 from the John Templeton Foundation.

## Author contributions

J.S. and N.D.dG. jointly conceived and designed this work. J.S. wrote an initial draft. J.S. and J.-P.A. analyzed the data. N.D.dG. provided resources. C.H. provided data and resources. J.S., N.D.dG., C.H., and J.-P.A. wrote the manuscript, and contributed to data analysis and discussions.

## Competing interests

The authors declare no competing interests.
