## [Transparent Peer Review File · Nature Communications]

The Three Stages of Religious Decline Around the World

Corresponding Author: Professor Jörg Stolz

Version 0:

Reviewer comments:

Reviewer #1

(Remarks to the Author)

Is there a worldwide secularization trend, and does it follow a particular pattern? This paper tackles this important question that has far-reaching significance. The researchers test the hypotheses that younger generations are less religious than older generations, which would suggest the arrow of time favours secularization, and whether there is a particular sequence of secularization, such that religious participation declines first, followed by the importance of religion, and finally by abandoning religious affiliation. They test these hypotheses with cross-country evidence (taking advantage of countries' different levels of secularization) as well as longitudinal data in the 40-year span of the WVS. While in recent years, there has been important research on secularization trends worldwide, the question of cohort effects and sequential patterns is currently not well understood. This paper, therefore, is a significant scientific advance and is of great interest to the social sciences.

This groundbreaking contribution will interest a wide range of scholars across many disciplines. However, I do have a few concerns that I hope the authors will address in a revision. Dealing with these issues will widen the potential impact of this important paper.

1) Religious belief (such as belief in God or the importance of God or Karma) was not in the religion variable and is not even discussed. This is baffling and weakens the paper considerably. The WVS has measures capturing religious belief, and it's hard to imagine any social science explanation of secularization that would entirely omit religious belief (for example, see Inglehart, 2021; Pew Research Center, 2015, The Global Religious Landscape; Voas & Chaves, 2016). It's not clear why the authors chose not to include religious belief (no explanation is given that I can find). If I were to speculate, based on the model they are testing and other evidence that belief and importance are highly related (e.g., Barber, 2012, CrossCult Res) I would expect that religious belief would be in the middle of the P-I-B sequence. I would like to see the authors address this omission and examine where religious belief falls in their model of secularization.

2) Including religious belief would be valuable because other theories of religious stability and decline posit that religious belief fades as a consequence of declines in credible displays of religious faith (see, for example, Henrich, 2009; EvHumBeh; Norenzayan et al., 2016; BehBrSci). This also aligns with the author's sequential model derived from Voas's sociological theory, but it posits a different psychological mechanism, credibility-enhancing displays or CREDs. In this alternative view, beliefs that are not backed up by public displays of faith are undermined and less likely to be culturally transmitted and stabilized (see, for example, J. Langston et al, 2018, RelBrBeh; Gervais & Najle, 2015, PsyRel&Spir).

3) The authors decided to exclude Eastern Europe and Israel from their main analyses, but I found their decision difficult to understand. Their main argument was that for historical reasons these two don't fit the pattern, but that is not a reason to just exclude them. Instead, why not include countries of the post-Communist space, create a dummy variable for them and show that the model fit improves significantly when they control for it, as other research papers have done? As the authors note, there are good explanations for why the post-Communist space, in particular, diverges from the patterns predicted by the model (it's even less clear to me why Israel was dropped). To the authors' credit, they do show this divergence in the Supplemental and discuss it, but I would recommend that they do not leave Eastern Europe out of their analytical models and have a more nuanced discussion in the main article. The omission of entire countries of Eastern Europe becomes particularly problematic as the authors seem to make a broad claim in several places (including in the abstract) that the cohort effect is applicable to the entire world and the march of secularization follows a specific sequence, followed by other statements that walk back this claim, such as "but other countries show only small, or even reversed cohort differences."

These statements should be moderated to make it clearer that the model's predictions fit the data in many countries and regions but certainly are not universal. It is still impressive how much the model explains, and it is not a weakness that there are some exceptions due to historical shocks and countervailing cultural circumstances.

4) The authors report that Muslim-majority countries fit the pattern predicted by the model, it's just that these countries are still in the very early stages of secularization. Yet the late Ronald Inglehart in his 2021 book and elsewhere report, using the same WVS data, that secularization is not discernible at all over time in these countries. I was curious how can we reconcile these discrepant findings?

5) The model presented here is not causal, although it is clearly derived from "modernization theory" that argues that religion declines with greater wealth, human development, education, and social safety nets. The authors presented a formal model in the Supplemental document section 1.A but did not present any analysis to test the model against the evidence. For example, doesn't this model predict that very high human development (or per capita GDP) should predict religious decline in all three aspects, moderate HD should predict declines in only the first and second, and low HD should predict decline in only religious participation?

6) Minor issues: Explain technical terms (such as "congregational religions.") Also the figure numbers in the main document are messed up and should be fixed.

(Remarks on code availability)

Reviewer #2

(Remarks to the Author)

The respecification of the secularization model is interesting.

The three measures show a strong factor. Does this factor structure change over time? In some ways, you would expect so if the model is correct.

The items themselves are problematic for non-Abrahamic religions. Buddhists or Hindus often have shrines at their home. It is not necessary for them to attend a house of worship. They may also attend those houses of worship for other reasons that are not religious.

How robust is the model when taking the secular response option (never attend religious services) as input for the calculations?

How distinct are the processes for participating and importance? Can you compute confidence or credibility intervals around the estimates?

How good is the fit for the longitudinal model? How was lambda chosen? If the lambda values are changed, does this affect fit?

You could replicate the cross-sectional analysis with different sets of the ESS/WVS from different waves.

What is the justification for taking the age of 40 as reference point? How sensitive are the models to the selection of different age reference points?

The analyses for the post communist countries (in Europe and Asia) could be repeated for different time intervals (taking into consideration the transition for each country).

Most importantly, what is the impact of HDI and education here? How well does this model hold up when considering the impact of HDI as a proxy for societal development that provides the replacement of organized religion.

(Remarks on code availability)

Reviewer #3

(Remarks to the Author)

Summary:

Using data from over 100 countries, the authors test a specification of secularization that religiosity declines in a specific order – participation, importance, then beliefs (P-I-B). The authors leave out Israel and some former Soviet Bloc countries but generally find that the model holds.

Comments:

This is pretty bold research. Up to this point, arguing that secularization is happening in countries that are modernizing has been somewhat challenging, though it appears as though the secularization argument may finally have been won. What

makes the P-I-B model bold is that it takes secularization as an assumption and then specifies the general order in which religiosity declines – participation first, then the importance of religion, followed by beliefs. I think the argument makes sense given that participation is the most “costly” aspect of religiosity and beliefs are the least “costly.” That the data generally support the argument is impressive.

The authors have also brought new statistical techniques to bear on this question. Adjusting for time in the way that they have in order to see whether the model works for countries at different levels of development is extremely innovative and compelling. I also appreciate having the code and data to replicate the work. We need to do more of that in the social sciences.

Perhaps one of the more noteworthy contributions of this study is that it is so amenable to future testing. The theory and resulting hypotheses are very clear – religiosity should follow a specific pattern of decline. This will allow scholars in countries around the world to analyze what is happening in their own countries to see if they align with the P-I-B model.

I was initially a little concerned by the authors exclusion of Israel and former Soviet Bloc, but I think they have provided enough of a justification in the article to defend that decision. Israel is unique in a lot of ways and it is not entirely clear why some former Soviet Bloc countries have seen a slight uptick in religious identification after the collapse of the USSR while others have become even more secular. Until we figure out why former Soviet Bloc countries are going the directions they are, I think it makes sense to exclude them.

Overall, I think this is an exciting paper. It makes a bold claim that is amenable to testing and will likely inspire lots of subsequent research. I know I will cite it regularly and may consider additional tests of this model.

(Remarks on code availability)

I happen to use R as my primary statistical analysis package. I looked over the code in detail. While I did not run the code (I could have), I'm familiar enough with R to understand what the authors did and how they did it. The code was clearly documented and made sense.

Version 1:

Reviewer comments:

Reviewer #1

(Remarks to the Author)

I read the cover letter and the revised manuscript. The authors have done a good job of addressing the reviewers' comments and criticisms, and I agree that this updated paper is now worthy of publication. I have no further issues. I look forward to seeing this paper in print.

(Remarks on code availability)

I recommend that as a standard "best practice," the authors commission an independent expert to review the code before this paper is published. If there are any errors, better to correct them pre-publication.

Reviewer #2

(Remarks to the Author)

Thank you for your revisions.

I have two main remaining comments at this stage.

1) You make a number of arguments on the impact of GDP and human development (HDI). It would be useful to explicitly correlate these indicators with the age gap measures. You refer to these when trying to explain anomalies in Africa. It would be important to explicitly report these associations for the whole data set to see whether GDP or HDI is indeed a possible correlate or predictor overall.

2) I think the longitudinal analysis is speculative. As you noted, there is quite a bit of misfit and it is not clear whether extrapolating from about 40 years of data can be scaled out to 200 years. We had a number of nationalistic wars, independence movements and major transitions in the last 200 years which are not well captured in the current data. I find this section unconvincing and it does distract from the larger argument.

(Remarks on code availability)

Reviewer #3

(Remarks to the Author)

Thank you for addressing my concerns. I have no further concerns with the paper. I believe it is an important and strong contribution to the study of secularization.

(Remarks on code availability)

Version 2:

Reviewer comments:

Reviewer #2

(Remarks to the Author)

Thank you for your revisions.

It would be useful to provide some relevant references when stating 'some scholars' on line 311.

I was struggling to understand Figure 2, especially the bottom panel. Could it be that the legend of the figure should have the religion components?

(Remarks on code availability)

Reviewer #4

(Remarks to the Author)

I was asked to review this manuscript with a focus on the empirical analysis. The manuscript seems competently executed and the overall empirical strategy is sound given the aims of the analysis. I have also reviewed the supplied code. Below I list a number of suggestions. They focus on more transparency regarding assumptions and sensitivity, (ii) more accurate reporting of technical details.

1/ The manuscript could be improved by providing a more detailed justification of the choice of function on p.6. I am not arguing that the logistic function is not a good choice. But it is essentially an arbitrary function form choice – even just out of the class of possible S-shaped curves (but of course it is not even clear why we should restrict our choices to these). Ideally, I would like to see a plot that is a version of Figure 2 with different function form choices so that a reader can gauge the variability in results stemming from different choices.

2/ I would like to see uncertainty indicated in Figure 2 and not the appendix. E.g., separate out the difference and plot credible intervals on both parts of the plot. If 95% CI make the figure hard to read, plot 50% intervals instead. This would still give the reader a good sense of the prediction uncertainty.

3/ The RHS variable “country secularity” is the first component from a principal component analysis. The uncertainty of these scores is not accounted for in the analysis, thus introducing errors-in-variables bias. This should at least be acknowledged in the discussion on p.19

4/ The LHS variable is a percentage difference calculated from surveys. The uncertainty of the difference is not accounted for in the model. While this matters less than for RHS variables, this should be acknowledged on p.19.

5/ The three outcomes (Participation, Importance, Belonging) are estimated jointly in the model code (i.e, the model likelihood consists of a system of three equations). Maybe this could be pointed out clearly in the Analytical strategy section.

6/ There is almost no detail on the MCMC sampler.

- The authors mention that they use Hamiltonian Monte Carlo, which would require specifying certain parameters (L, epsilon). Inspection of the code shows that they use the NUTS sampler by Gelman and Hoffman (JMLR 2014), which chooses these parameters automatically.
- Inspection of the code shows that results are based on 5000 MCMC sampler.
- There is no discussion of diagnostics for the absence of MCMC convergence
- There is no discussion of a robustness / perturbation / sensitivity analysis to different prior choices (see chapter 6.2. of Gill J., 2008, Bayesian Methods, for a helpful discussion of prior sensitivity analyses)
- There is no discussion of the inspection of divergent transitions, which is an important issue that arises in this type of Hamiltonian MCMC.

7/ Why is one set of models implemented in a Bayesian framework and the other (on p.14) in a frequentist framework? At a minimum this should be pointed out clearly in the methodology section.

8/ The appendix tables should state what posterior quantities they report. I assume they are posterior means and standard deviations.

9/ The ms provides little to no detail about the multiple imputation procedure used. It uses chained equations, but it does not specify the imputation method nor does it list the number of imputed data sets created. Inspection of the code shows that predictive mean matching is used (i.e., no accounting for time trends in the imputation) and that M=5 data sets are created. Note that 5 imputed data sets is very low for some quantities of interest such 2.5, 97.5 quantiles of the posterior distribution.

(Remarks on code availability)

I have reviewed the code. It is clear enough for the reader to follow along. There is a README file listing detailed instructions and software requirements (small correction: the file lists Rstudio as a requirement. This is of course not needed. Proper R is enough).

Suggestions:

Instead of instructions of the type "run lines 1-11 in file 1, then lines 3-1111 in file 2" why not simply write a Makefile?

The code as provided now does not replicate cleanly. As an example, I attach a simple TEST.R file that follows the readme: it sets some choice variables (which are used in ifelse statements to select model types) and then sources Sscript/B_main.R. The output in TEST.ROUT shows error msgs.

Version 3:

Reviewer comments:

Reviewer #4

(Remarks to the Author)

The authors did an excellent job addressing the issues raised in the review. The responses are clear and sensible and the added text increases (in my view) the clarity and precision of the manuscript.

I support the publication of the manuscript.

(Remarks on code availability)

Reactions to reviews of “How religion fades: The three stages of religious decline around the world”

We thank the reviewers for their thoughtful comments. We have revised the paper considering their suggestions and believe that the paper has been considerably strengthened by the revision. In what follows, we react to all points raised and show how we adjusted the manuscript.

The most important changes are the following:

(1) We now discuss the place of belief in our scheme, as suggested by reviewer 1. We argue that importance of religion is a proxy for beliefs and show in the Appendix (part 9) that different composite measures of “personal religiosity” (including belief) behave similarly to importance of religion. We also include a paragraph on the relationship between our model and the theoretical idea of CRED’s.

(2) Whereas the previous version excluded eastern post-communist countries and Israel, we now include all countries in our analysis, adding a dummy variable for post-communist countries in the controls (a point raised by reviewer 1 and reviewer 3). We therefore added Figure 5 to the main text where Western Post-communist and Eastern Post-communist countries are compared.

(3) We have strengthened the claim of the paper. We now claim that the P-I-B sequence is discernible not just in historically Christian, but also Buddhist/Hindu and Muslim countries. This significant strengthening of the paper was made possible by two new developments. For one thing, Pew Research Center just came out with new data on several Asian countries. We have included these results which raises the number of countries in the Pew dataset to 111. Seven countries are added to the dataset (Malaysia, Cambodia, Hong Kong, Singapore, Sri Lanka, Taiwan, Thailand). Previously, we relied on Pew data for India, Indonesia, Japan, South Korea, and Vietnam that were collected in Pew’s Global Attitudes surveys that were in some cases missing at least one measure. Since Pew has recently concluded religion-focused surveys in each country, our new dataset uses the more recent, complete set of measures now available in these countries. In the previous version we had so few Asian countries that little could be said about the usefulness of our model in traditionally Buddhist and Hindu countries. Now, however, it turns out that our model describes the secularization process in these countries as well. This also permits to counter possible criticism that our indicators might not perform well for non-congregational (e.g., Hindu, Buddhist) religions. For another thing, it occurred to us that we could test the P-I-B sequence also in historically Muslim countries separately. While in historically Muslim countries and in African countries the last step of the sequence is absent, we do find the first two steps. This is compatible with the model if historically Muslim countries are in an early phase of the secular transition.

In what follows, we react to the specific queries and suggestions of the reviewers:

Reviewer #1

Is there a worldwide secularization trend, and does it follow a particular pattern? This paper tackles this important question that has far-reaching significance. The researchers test the hypotheses that younger generations are less religious than older generations, which would suggest the arrow of time favours secularization, and whether there is a particular sequence of

secularization, such that religious participation declines first, followed by the importance of religion, and finally by abandoning religious affiliation. They test these hypotheses with cross-country evidence (taking advantage of countries' different levels of secularization) as well as longitudinal data in the 40-year span of the WVS. While in recent years, there has been important research on secularization trends worldwide, the question of cohort effects and sequential patterns is currently not well understood. This paper, therefore, is a significant scientific advance and is of great interest to the social sciences.

This groundbreaking contribution will interest a wide range of scholars across many disciplines. However, I do have a few concerns that I hope the authors will address in a revision. Dealing with these issues will widen the potential impact of this important paper. Thank you.

1) Religious belief (such as belief in God or the importance of God or Karma) was not in the religion variable and is not even discussed. This is baffling and weakens the paper considerably. The WVS has measures capturing religious belief, and it's hard to imagine any social science explanation of secularization that would entirely omit religious belief (for example, see Inglehart, 2021; Pew Research Center, 2015, *The Global Religious Landscape*; Voas & Chaves, 2016). It's not clear why the authors chose not to include religious belief (no explanation is given that I can find). If I were to speculate, based on the model they are testing and other evidence that belief and importance are highly related (e.g., Barber, 2012, *CrossCult Res*) I would expect that religious belief would be in the middle of the P-I-B sequence. I would like to see the authors address this omission and examine where religious belief falls in their model of secularization.

We agree with reviewer 1 that belief is an important dimension of religiosity. However, belief indicators vary strongly with religion and are more problematic for global comparisons than other indicators of religiosity. As reviewer 1 correctly suspects, belief variables are often strongly correlated with perceived importance of religion, which is why we use an importance measure as a proxy for belief. However, in additional analyses we now show that using composite measures including belief (such as "belief in god" or "importance of god") give similar results. We have made this clearer at different points in the text and have added the following paragraph:

What we measure with "importance of religion in one's life" is an indicator for personal religiosity in a very broad sense including religious beliefs and personal practices such as prayer. These different elements, and especially belief variables vary strongly across different religious and cultural areas. We have therefore chosen to use only one indicator both in the Pew study and the WVS/EVS data. Its meaning seems to be fairly comparable across cultures and it clearly taps personal religiosity. As a robustness check, we investigate how well our results hold when measuring the "importance" dimension in two other ways: (a) with a composite measure of importance of religion and prayer; and (b) with a composite measure of importance of religion, importance of god, belief in god, self-description as a religious person, and confidence in religious institutions (online Appendix part 9).

2) Including religious belief would be valuable because other theories of religious stability and decline posit that religious belief fades as a consequence of declines in credible displays of religious faith (see, for example, Henrich, 2009; *EvHumBeh*; Norenzayan et al., 2016; *BehBrSci*). This also aligns with the author's sequential model derived from Voas's sociological theory, but it posits a different psychological mechanism, credibility-enhancing

displays or CREDs. In this alternative view, beliefs that are not backed up by public displays of faith are undermined and less likely to be culturally transmitted and stabilized (see, for example, J. Langston et al, 2018, *RelBrBeh*; Gervais & Najle, 2015, *PsyRel&Spir*).

This is an interesting point. We have added the following paragraph in the main paper on how the evolutionary anthropology account of CRED's may specify our theoretical story, citing the literature suggested by reviewer 1.

Such an account is compatible and may be specified with ideas of evolutionary anthropology involving CRED's.¹⁷ The central idea of CRED's is that individuals will believe and imitate others with a higher probability if these others back up their words with hard-to-fake deeds¹⁸. Especially in religion, where supernatural entities are notoriously invisible, CRED's are thought to be of primary importance^{19,20}. Because of the increase and increasing efficiency of secular goods, parents may still teach their children religiosity - but stop backing it up with religious behavior. For example, parents may teach their children that they have to go to the religious ritual (or send them to institutions where such attendance is obligatory), but parent's fail to often attend themselves. Children will notice this, find this religiosity decreasingly convincing and drop their attendance even lower. Likewise, parents may teach their children that religion is important - but fail to show such an importance in their daily lives. Children may then perceive the importance of religion as even less.

3) The authors decided to exclude Eastern Europe and Israel from their main analyses, but I found their decision difficult to understand. Their main argument was that for historical reasons these two don't fit the pattern, but that is not a reason to just exclude them. Instead, why not include countries of the post-Communist space, create a dummy variable for them and show that the model fit improves significantly when they control for it, as other research papers have done? As the authors note, there are good explanations for why the post-Communist space, in particular, diverges from the patterns predicted by the model (it's even less clear to me why Israel was dropped). To the authors' credit, they do show this divergence in the Supplemental and discuss it, but I would recommend that they do not leave Eastern Europe out of their analytical models and have a more nuanced discussion in the main article. The omission of entire countries of Eastern Europe becomes particularly problematic as the authors seem to make a broad claim in several places (including in the abstract) that the cohort effect is applicable to the entire world and the march of secularization follows a specific sequence, followed by other statements that walk back this claim, such as "but other countries show only small, or even reversed cohort differences." These statements should be moderated to make it clearer that the model's predictions fit the data in many countries and regions but certainly are not universal. It is still impressive how much the model explains, and it is not a weakness that there are some exceptions due to historical shocks and countervailing cultural circumstances.

We follow the recommendation of reviewer 1 to include all countries in the models. We therefore have made the following changes: We include all countries in our Bayesian models and use a single dummy variable indicating all eastern post-communist countries. For our figures, we proceed as follows: We include Israel in all figures and models, explaining it as an outlier in the text. For better visibility of the overall pattern, we exclude eastern post-communist countries from Fig. 1, 3, 4, and 7, noting this in the legend. In contrast, we have moved the description and explanation of the specificity of eastern post-communist countries from the online Appendix to the main text. This includes what is now Fig. 5., showing the comparison of PIB in western and eastern post-communist countries.

4) The authors report that Muslim-majority countries fit the pattern predicted by the model, it's just that these countries are still in the very early stages of secularization. Yet the late Ronald Inglehart in his 2021 book and elsewhere report, using the same WVS data, that secularization is not discernible at all over time in these countries. I was curious how can we reconcile these discrepant findings?

We have added the following footnote:

“In contrast to Inglehart (2021), we do see signs of secularization in the Muslim world both in the Pew and the EVS/WVS data. Inglehart looks at differences with regard to all possible indicators and reports small or no differences between old and young cohorts. Our model however expects such differences only with regard to participation - where they do in fact appear.”

5) The model presented here is not causal, although it is clearly derived from “modernization theory” that argues that religion declines with greater wealth, human development, education, and social safety nets. The authors presented a formal model in the Supplemental document section 1.A but did not present any analysis to test the model against the evidence. For example, doesn't this model predict that very high human development (or per capita GDP) should predict religious decline in all three aspects, moderate HD should predict declines in only the first and second, and low HD should predict decline in only religious participation?

Yes, reviewer 1 is right in that the formal model goes further than what we claim in our main paper and makes causal predictions. This formal model presents the theoretical background of our demonstration (a form of secularization theory). Testing the causal model, however, would necessitate a new paper (we are working on this). One of the problems one faces is how to measure the secular goods that stand in competition with religious goods. In many studies HDI or some similar measures are used, but there are many reasons why this may be unsatisfactory. We have added this point in the online Appendix, part 1.

6) Minor issues: Explain technical terms (such as “congregational religions.”) Also the figure numbers in the main document are messed up and should be fixed.

Done

Reviewer #2

The respecification of the secularization model is interesting.

Thank you.

The three measures show a strong factor. Does this factor structure change over time? In some ways, you would expect so if the model is correct.

If reviewer 2 alludes to questions of measurement invariance, we agree. The model argues that all three indicators lie on one “dimension”. Since there is a consecutive dropping over time, the “factor structure” or “correlational structure” cannot remain constant. We have added the following paragraph:

On the subject of measurement invariance, our model makes an additional point. Studies that test measurement invariance for religiosity items over a large number of countries and that assume all countries to be “at the same time” may get it wrong. They may conclude that

there is measurement invariance, while, in fact, countries are at different points in time on the secular transition and therefore have to show different correlational structures.

The items themselves are problematic for non-Abrahamic religious. Buddhists or Hindus often have shrines at their home. It is not necessary for them to attend a house of worship. They may also attend those houses of worship for other reasons that are not religious. Yes, we agree. This is an inherent limitation of our data, as we point out in the methods section. We have slightly rewritten the text to further point out this limitation.

How robust is the model when taking the secular response option (never attend religious services) as input for the calculations?

The “never attend religious services” response option is always used as it is one of the values of the variable itself. Our current results include this.

How distinct are the processes for participating and importance? Can you compute confidence or credibility intervals around the estimates?

Yes, we calculate credibility (highest (posterior) density) intervals around our estimates. These are visualized in Fig. 6. The estimates for participating and importance are significantly different in all three datasets. We have now also added the specific statistical information in the main text.

How good is the fit for the longitudinal model? How was lambda chosen? If the lambda values are changed, does this affect fit?

As we show in the paper, the fit is substantial, but obviously far from perfect. There seem to be several country-specific factors that equally influence the secular transition and are added to the overall process. In the online appendix, part 2 we explain how we calculated lambda. Intuitively, it is the value that maximizes the fit of a sigmoid function to the different country regression lines, when “stretching out theoretical time”.

You could replicate the cross-sectional analysis with different sets of the ESS/WVS from different waves.

Yes, this is what we do in the section “Replicating the P-I-B sequence longitudinally” of the paper.

What is the justification for taking the age of 40 as reference point? How sensitive are the models to the selection of different age reference points?

We use the age of 40 for comparability as this is the cut-off point already used in the original Pew analysis. In part 10 of the online appendix, we show that our results are robust to changing the cut-off point for cohort.

The analyses for the post communist countries (in Europe and Asia) could be repeated for different time intervals (taking into consideration the transition for each country).

This suggestion reflects the point raised by reviewer 1. We have now moved the analysis of post-communist countries from the online appendix to the main text.

Most importantly, what is the impact of HDI and education here? How well does this model hold up when considering the impact of HDI as a proxy for societal development that provides the replacement of organized religion.

This question is similar to the point raised by reviewer 1 regarding our formal model (where a classical secularization theory is implied). As mentioned above, such a model lies in the

background of our demonstration, but we do not aim to test it in this paper. Doing so would necessitate a different paper.

Reviewer #3

Summary:

Using data from over 100 countries, the authors test a specification of secularization that religiosity declines in a specific order – participation, importance, then beliefs (P-I-B). The authors leave out Israel and some former Soviet Bloc countries but generally find that the model holds.

Comments:

This is pretty bold research. Up to this point, arguing that secularization is happening in countries that are modernizing has been somewhat challenging, though it appears as though the secularization argument may finally have been won. What makes the P-I-B model bold is that it takes secularization as an assumption and then specifies the general order in which religiosity declines – participation first, then the importance of religion, followed by beliefs. I think the argument makes sense given that participation is the most “costly” aspect of religiosity and beliefs are the least “costly.” That the data generally support the argument is impressive.

Thank you

The authors have also brought new statistical techniques to bear on this question. Adjusting for time in the way that they have in order to see whether the model works for countries at different levels of development is extremely innovative and compelling. I also appreciate having the code and data to replicate the work. We need to do more of that in the social sciences. Perhaps one of the more noteworthy contributions of this study is that it is so amenable to future testing. The theory and resulting hypotheses are very clear – religiosity should follow a specific pattern of decline. This will allow scholars in countries around the world to analyze what is happening in their own countries to see if they align with the P-I-B model.

Thank you

I was initially a little concerned by the authors exclusion of Israel and former Soviet Bloc, but I think they have provided enough of a justification in the article to defend that decision. Israel is unique in a lot of ways and it is not entirely clear why some former Soviet Bloc countries have seen a slight uptick in religious identification after the collapse of the USSR while others have become even more secular. Until we figure out why former Soviet Bloc countries are going the directions they are, I think it makes sense to exclude them.

This point has been raised by both reviewers 1 and 2. We have now included Israel and moved the discussion of the post-communist countries from the online appendix to the main text

Overall, I think this is an exciting paper. It makes a bold claim that is amenable to testing and will likely inspire lots of subsequent research. I know I will cite it regularly and may consider additional tests of this model.

Thank you for these points. Reviewer 3 has made us aware of this strength. As a consequence, we have slightly rewritten the part on strengths in the conclusion. We write:

Our findings do not depend on the Pew Data with which our main results are demonstrated. Both a cross-sectional and longitudinal analysis of WVS/EVS data show exactly the same three steps hypothesized by our theory. Finally, our model makes bold predictions that may be tested in future work.

Reviewer #3 (Remarks on code availability):

I happen to use R as my primary statistical analysis package. I looked over the code in detail. While I did not run the code (I could have), I'm familiar enough with R to understand what the authors did and how they did it. The code was clearly documented and made sense.
Thank you

Reactions to reviews of “How religion fades: The three stages of religious decline around the world”. Second round

Jörg Stolz, Nan Dirk de Graaf, Conrad Hackett, Jean-Philippe Antonietti

We are delighted that two reviewers think the manuscript is publishable as it is. We thank reviewer 1 for the suggestion of an additional independent code-check and reviewer 2 for the two remaining points.

This text explains how we have

1. addressed all comments by the reviewers and the external expert checking the code.
2. added information both in the main text and the Online Appendix on gender, to comply with the ‘Sex and Gender Equity in Research – SAGER – guidelines’.
3. added two very minor further adjustments.

We have updated the following checklists:

- Editorial policy checklist:
- Reporting summary

The revised replication package is available at OSF:

<https://osf.io/vczta/>

1. Comments by reviewers

Reviewer #1 (Remarks to the Author):

I read the cover letter and the revised manuscript. The authors have done a good job of addressing the reviewers' comments and criticisms, and I agree that this updated paper is now worthy of publication. I have no further issues. I look forward to seeing this paper in print.

Thank you.

Reviewer #1 (Remarks on code availability):

I recommend that as a standard "best practice," the authors commission an independent expert to review the code before this paper is published. If there are any errors, better to correct them pre-publication.

Thank you for this suggestion. We have commissioned an external independent expert, Dr. Simon Brauer, Research area specialist at the Institute for Social Research, University of Michigan, who has made a 20h-long in-depth code-check. Simon Brauer is not only an expert R programmer but also has domain knowledge in the sociology of religion. He has found seven minor issues in the script (e.g., one issue with an erroneous plot labelling, one issue concerning the type of error bar, two issues that may arise when replicating the results and that may reverse the sign of a shifter, one issue where three countries were assigned to the wrong continent (but did not change the output), two issues that arise with the version of R). We have fixed these issues in the new version of the script. There were no issues that substantively changed the results.

Reviewer #2 (Remarks to the Author):

Thank you for your revisions. I have two main remaining comments at this stage.

1) You make a number of arguments on the impact of GDP and human development (HDI). It would be useful to explicitly correlate these indicators with the age gap measures. You refer to these when trying to explain anomalies in Africa. It would be important to explicitly report these associations for the whole data set to see whether GDP or HDI is indeed a possible correlate or predictor overall.

Thank you for this remark that has made us aware that at one point in the manuscript we made a causal claim about the relationship of HDI and the beginning of the secular transition. This claim was not backed up by data.

Our paper seeks to show that increasing country secularity coincides with the P-I-B sequence. It does not attempt to causally explain country secularity - this would mean writing a completely different paper. We have therefore deleted the reference to HDI at that point in the manuscript (p. 6). However, since the question of explanation is of course an important one, we have added a new paragraph in the conclusion, reading as follows:

Our study has demonstrated that increasing country secularity coincides with the P-I-B sequence. By doing so, we have sidestepped the question of just what non-religious factors causally explain country secularity in the first place. What necessary and sufficient conditions may lead a country to enter the secular transition and to keep it moving along the transition trajectory? While it is obvious that indicators of modernization, such as HDI, are linked to the level of country secularity^{1, 2, 3} and to the religiosity-cohort gaps on an aggregate level (Appendix 14), the underlying causal mechanisms have yet to be disentangled.

In the Online Appendix A14, we give the correlations and visualizations of overall aggregated country secularity, HDI and GDP/capita. We also visualize the countries' P-I-B cohort differences according HDI for the Pew data. These results show that there is evidently a very strong relationship between HDI, country secularity, and the age gap measures. A paper attempting to causally explain the level of cohort gaps would however still have to be written.

2) I think the longitudinal analysis is speculative. As you noted, there is quite a bit of misfit and it is not clear whether extrapolating from about 40 years of data can be scaled out to 200 years. We had a number of nationalistic wars, independence movements and major transitions in the last 200 years which are not well captured in the current data. I find this section unconvincing and it does distract from the larger argument.

We agree with reviewer 2 that the assumptions made in the longitudinal analysis are relatively strong (as in the original theory by Voas) and that this point might be made clearer in the paper. More generally, it is true that events such as nationalistic wars or independence movements leading to religious revivals are not apparent in the data (except, very clearly, for the Orthodox resurgence). But this is actually what the model assumes: that many short-term events will have only slight or no influences on the long-term three-stage pattern of secular transition.

To react to this point, we have rewritten the interpretations of the longitudinal analysis (in the data analysis section, p. 15, and the conclusion, p. 17) highlighting the strong assumptions underlying this analysis. We argue that the interpretation of the longitudinal results require caution. Note that, in our rewrite, we have not used the term "speculative". In our view, speculative would mean making just guesses about possible outcomes. In our case, however, we test and confirm a hypothesis that has been deduced from a theory. Furthermore, we analyze the richest dataset currently available for our purpose; therefore, this is the best test of our theory with longitudinal data.

We write in the main interpretation of the results:

The results of our longitudinal analysis offer support for the main hypothesis but should be approached with caution. First, the model does not adequately fit the data for some countries. Second, the findings rest on the assumption that the average pace of the secular transition remains constant—an assumption that may not hold true. Lastly, the analysis assumes that data from a forty-year period can be extended to predict trends over more than two centuries, including the assumption that, once begun, the secular transition would remain unaffected by phenomena such as changes in state policies, nationalistic conflicts, or independence movements. That having been said, we analyze the richest dataset currently available for our purpose; therefore, this is the best test of our theory with longitudinal data. (p. 15)

and in the conclusion:

A final and arguably the most important point is the limited temporal scope of the longitudinal data, which warrants caution. We are assuming that these 40 years, across different countries, reflect distinct stages of a shared trajectory spanning over 200 years. This is a significant assumption that could ultimately prove inaccurate. (p. 17)

Reviewer #3 (Remarks to the Author):

Thank you for addressing my concerns. I have no further concerns with the paper. I believe it is an important and strong contribution to the study of secularization.

Thank you.

2. Additional text and analyses on gender

In our previous version, gender had already been used as a control variable in the We have sought to comply more fully with the ‘Sex and Gender Equity in Research – SAGER – guidelines’.⁴ We have introduced the following text:

Main text, introduction:

We use several control variables. One of them is gender (self-reported), which is introduced via gender ratio in our models. We also compute the P-I-B sequence separately for men and women.

Main text, results section:

The replication with WVS/EVS 7 permits checking the whether the P-I-B sequence is robust to the control of gender ratio and separately for women and men (Online Appendix 12 and 13).

Main text conclusion section:

The P-I-B sequence can furthermore be demonstrated both for women and men.

Main text, methods section:

Gender. We use both the (self-reported) dichotomous variable gender (male/female), and the gender ratio expressed as the percentage of women in the respective population or subpopulation. A large literature shows that women are more religious than men in western countries, even when controlling for various socio-structural variables^{50, 51}. A world-wide assessment, however, suggests that such a gender gap is mainly present in societies with Christian majorities^{20, 52}. In our models, we use gender ratio as a control variable. See online Appendix 12 where it is shown that the P-I-B is robust to including or excluding the control variable gender ratio (and other control variables). See furthermore online Appendix 13, where the P-I-B sequence is calculated separately for women and men.

In the Online Appendix 13 we give the P-I-B results for WVS/EVS7 data separately for women and men.

3. Additional minor adjustments

(1) We have changed the wording for the Notes for Figures 3-5 as follows:

We plot the 95% confidence intervals. For better visibility, only half of the confidence interval is plotted - for positive percentage values at the right of the respective percentage bar, for negative percentage values at the left of the respective percentage bar.

(2) We have added the following paragraph on p. 9.

We acknowledge that participation in acts of worship at temples within Asian religions differs significantly from participation in congregational religions. For this reason, we employed a specialized measure tailored to East Asian societies in order to accurately assess participation in that region (see the methods section under "dependent variables" below). Notably, when using data specifically designed to measure participation at Buddhist sites, we observe the same age-related patterns in attendance that are evident in religions emphasizing congregational worship. This is particularly remarkable given that some scholars have argued that concepts such as "participation" or "belonging" are not applicable to non-congregational Asian religions such as Buddhism or Hinduism.

Literature

1. Inglehart RF. *Religion's Sudden Decline. What's Causing it, and What Comes Next?* Oxford University Press (2021).
2. Molteni F. *A Need for Religion. Insecurity and Religiosity in the Contemporary World.* Brill (2021).
3. Hackett C, Kramer S, Fahmy D, Marshall J. *The Age Gap in Religion Around the World*, <https://www.pewforum.org/2018/06/13/the-age-gap-in-religion-around-the-world/> edn. Pew, Research Center (2018).

4. Heidari S, Babor TF, Sera Tort PDC, Curno M. Sex and Gender Equity in Research: rationale for the SAGER guidelines and recommended use. *Research Integrity and Peer Review* **1**, (2016).

Reactions to reviews of “How religion fades: The three stages of religious decline around the world”. Third round

We thank Reviewer 4 for the expert look at the empirical analysis and the suggestions how transparency regarding assumptions and reporting of technical details could be improved. We were, of course, happy with Reviewer 4's general assessment that "The manuscript seems competently executed and the overall empirical strategy is sound given the aims of the analysis." We thank Reviewer 2 for having spotted two remaining issues. We believe that the revision has again strengthened the paper, especially on reporting of technical details and ease of reproducibility.

We react to all points raised as follows.

Reviewer 2

Provide some relevant references when stating 'some scholars' on line 311. (Reviewer 2)
Yes, we have added three references.

I was struggling to understand Figure 2, especially the bottom panel. Could it be that the legend of the figure should have the religion components? (Reviewer 2)
Yes, thank you for spotting this! We have corrected the legend.

Reviewer 4

I was asked to review this manuscript with a focus on the empirical analysis. The manuscript seems competently executed and the overall empirical strategy is sound given the aims of the analysis.
Thank you

I have also reviewed the supplied code. Below I list a number of suggestions. They focus on more transparency regarding assumptions and sensitivity, (ii) more accurate reporting of technical details.

1/ The manuscript could be improved by providing a more detailed justification of the choice of function on p.6. I am not arguing that the logistic function is not a good choice. But it is essentially an arbitrary function form choice – even just out of the class of possible S-shaped curves (but of course it is not even clear why we should restrict our choices to these). Ideally, I would like to see a plot that is a version of Figure 2 with different function form choices so that a reader can gauge the variability in results stemming from different choices.

We agree with Reviewer 4 that alternative functional forms could have been used. We now acknowledge this choice in the main text and show in Appendix part 14 what our results would graphically look like with a Probit function (Figure 6). The results are very similar. In the main text, we write:

We model our three dependent variables using a sigmoid (inverse logit) function. We chose the logit function due to its intuitive interpretability compared to alternative link functions. However, one might question whether the results would differ under alternative specifications. A good candidate would be the probit function. Non-parametric functions are not feasible because of the relatively small number of countries. In Appendix part 14 we show

graphically that results look very similar with probit. AIC measures of logit and probit are also extremely close.

2/ I would like to see uncertainty indicated in Figure 2 and not the appendix. E.g., separate out the difference and plot credible intervals on both parts of the plot. If 95% CI make the figure hard to read, plot 50% intervals instead. This would still give the reader a good sense of the prediction uncertainty.

Thank you for this suggestion. We have plotted credible intervals for all curves in Figure 2. Following the suggestion of Reviewer 4, and to increase readability, we have plotted 50% CI instead of 95% CI.

3/ The RHS variable “country secularity” is the first component from a principal component analysis. The uncertainty of these scores is not accounted for in the analysis, thus introducing errors-in-variables bias. This should at least be acknowledged in the discussion on p.19

We have added the following paragraph in the analytic strategy section (p. 19). This paragraph also incorporates the next point 4/

We acknowledge that our analysis does not explicitly incorporate uncertainty in the scores of our dependent variables participation, importance, and belonging, as well as our independent variable, country secularity. While we considered addressing this issue using latent structural equation modeling, we ultimately opted against it to maintain the tractability of an already complex model. However, robustness checks (Appendix 8,9,10,11) indicate that our findings are consistent across different ways of measuring and specifying our dependent and independent variables.

4/ The LHS variable is a percentage difference calculated from surveys. The uncertainty of the difference is not accounted for in the model. While this matters less than for RHS variables, this should be acknowledged on p.19.

Yes, we now acknowledge this in the main text (see previous point).

5/ The three outcomes (Participation, Importance, Belonging) are estimated jointly in the model code (i.e, the model likelihood consists of a system of three equations). Maybe this could be pointed out clearly in the Analytical strategy section.

This is an excellent point! We now indicate this in the Analytical strategy section with the following sentence:

The three outcomes are estimated jointly within a system of three equations that together define the model likelihood.

6/ There is almost no detail on the MCMC sampler.

We have clarified our use of the MCMC sampler in the following way:

- The authors mention that they use Hamiltonian Monte Carlo, which would require specifying certain parameters (L, epsilon). Inspection of the code shows that they use the NUTS sampler by Gelman and Hoffman (JMLR 2014), which chooses these parameters automatically.

- Inspection of the code shows that results are based on 5000 MCMC sampler.

Correct, we now give this information in the main text.

- There is no discussion of diagnostics for the absence of MCMC convergence

Correct, we now give this information in the main text. The MCMC sampler converges.

- There is no discussion of a robustness / perturbation / sensitivity analysis to different prior choices (see chapter 6.2. of Gill J., 2008, Bayesian Methods, for a helpful discussion of prior sensitivity analyses)

We now perform a robustness test by doubling the variance. We cite Gill.

- There is no discussion of the inspection of divergent transitions, which is an important issue that arises in this type of Hamiltonian MCMC.

We now give information on the fit diagnostics. There are no divergent transitions.

In the main text, this now reads as follows:

We employ a Hamiltonian Monte Carlo (HMC) algorithm with four chains and 5,000 iterations, utilizing the No-U-Turn Sampler (NUTS) developed by Hoffman and Gelman¹. The models converge. The fit diagnostics show satisfactory treedepth, satisfactory E-BFMI, and satisfactory effective sample size. There are no divergent transitions. We performed a local sensitivity analysis to evaluate the robustness of our findings to different prior specifications. Following Gill², we explored less informative priors. As further reducing the degrees of freedom was not advisable, we instead doubled the variance while keeping the degrees of freedom fixed and assessed the impact on the posterior distribution. The results indicate that this adjustment has a negligible effect on the posterior parameters (Appendix part 15).

7/ Why is one set of models implemented in a Bayesian framework and the other (on p.14) in a frequentist framework? At a minimum this should be pointed out clearly in the methodology section.

Yes, we do indeed combine a Bayesian and a frequentist framework.

We now clarify this as follows:

In this paper, we combine a frequentist and Bayesian approach. The confidence intervals in Figures 1,3,4, and 5 follow a frequentist logic (they are produced to comply with the journal's statistical guidelines). All other inferences use a Bayesian logic. We follow Gelman and Shalizi in their view that both frequentist and Bayesian methods can be combined pragmatically in a hypothetico-deductive scientific framework.³ We mainly use Bayesian methods because the relatively complex functions needed a flexible estimation approach. We believe that such a combination of frequentist and Bayesian methods is defensible both on philosophical³ and pragmatic⁴ grounds.

8/ The appendix tables should state what posterior quantities they report. I assume they are posterior means and standard deviations.

We report posterior midpoints and standard errors. We have added this information where it was missing in the Notes of Tables 4, 5, 8, and 9.

9/ The ms provides little to no detail about the multiple imputation procedure used. It uses chained equations, but it does not specify the imputation method, nor does it list the number of imputed data sets created. Inspection of the code shows that predictive mean matching is used (i.e., no accounting for time trends in the imputation) and that M=5 data sets are created. Note that 5 imputed data sets is very low for some quantities of interest such 2.5, 97.5 quantiles of the posterior distribution.

We now impute 20 datasets and clarify our procedure of imputing missing values as follows:

In the Pew and WVS/EVS 7 datasets we imputed missing values with predictive mean matching (method = "pmm") using the mice library (multivariate imputation by chained equations). We used 10 imputed dataset (m = 10). In the WVS1-7, we imputed missing values

of the variables participation, belonging, and confidence in the church (all three with 2% of missing values) with a LOCF/NOCB (forward/backward fill) imputation procedure inside the countries and sorting for year. We imputed missing values in the variable importance of religion (with 9.8% of missing values) with a two-level Bayesian linear mixed model approach (mice package, method = "2lpan", predictors: confidence in church, participation, belonging). The control variables discrimination, legislation, and regulation had 21.6% of missing variables. These values were imputed with LOCF/NOCB, since we did not have evident predictors for a "2lpan" approach. In a robustness test (Appendix Table 8) we show that imputing or not imputing missing values does not change the results substantively.

Reviewer #4 (Remarks on code availability):

I have reviewed the code. It is clear enough for the reader to follow along.

Thank you.

There is a README file listing detailed instructions and software requirements (small correction: the file lists Rstudio as a requirement. This is of course not needed. Proper R is enough).

We have deleted Rstudio as a requirement.

Suggestions:

Instead of instructions of the type "run lines 1-11 in file 1, then lines 3-1111 in file 2" why not simply write a Makefile?

We have rewritten the top-level script to make the running of both the demo and the overall script very easy.

The code as provided now does not replicate cleanly. As an example, I attach a simple TEST.R file that follows the readme: it sets some choice variables (which are used in ifelse statements to select model types) and then sources Script/B_main.R. The output in TEST.ROUT shows error msgs.

To ensure better reproducibility, we have rewritten the Readme file and given instructions on how to set up the environment (we suspect that a different environment may have caused the replication problems that Reviewer 4 encountered). We have also simplified the organization of the overall script and have asked two colleagues to download and run the replication package from OSF. The script should now replicate cleanly.

Lausanne, 27.3.2025

Jörg Stolz, Nan Dirk de Graaf, Conrad Hackett, Jean-Philippe Antonietti

References

1. Hoffman MD, Gelman A. The No-U-Turn Sampler: Adaptively Setting Path Lengths in Hamiltonian Monte Carlo. *Journal of Machine Learning Research* **15**, 1593-1623 (2014).
2. Gill J. *Bayesian Methods. A Social and Behavioral Sciences Approach. Third Edition.* Taylor and Francis (2015).

3. Gelman A, Shalizi CR. Philosophy and the practice of Bayesian statistics. *British Journal of Mathematical and Statistical Psychology* **66**, 8-38 (2013).
4. Bayarri MJ, Berger JO. The interplay of Bayesian and frequentist analysis. *Statistical Science* **19**, 58-80 (2004).